# SYNTHESIZING HIGH-QUALITY VISUAL QUESTION ANSWERING FROM MEDICAL DOCUMENTS WITH GENERATOR-VERIFIER LMMS

**Xiaoke Huang**[1*]**, Ningsen Wang**[1,2*]**, Hui Liu**[3]**, Xianfeng Tang**[3]**, Yuyin Zhou**[1]

[1] UC Santa Cruz      [2] Fudan University      [3] Amazon Research

* indicates equal contribution

https://github.com/UCSC-VLAA/MedVLSynther

## ABSTRACT

Large Multimodal Models (LMMs) are increasingly capable of answering medical questions that require joint reasoning over images and text, yet training general medical VQA systems is impeded by the lack of large, openly usable, high-quality corpora. We present **MedVLSynther**, a rubric-guided generator-verifier framework that synthesizes high-quality multiple-choice VQA items directly from open biomedical literature by conditioning on figures, captions, and in-text references. The generator produces self-contained stems and parallel, mutually exclusive options under a machine-checkable JSON schema; a multi-stage verifier enforces essential gates (self-containment, single correct answer, clinical validity, image-text consistency), awards fine-grained positive points, and penalizes common failure modes before acceptance. Applying this pipeline to PubMed Central yields *MedSynVQA*: 13,087 audited questions over 14,803 images spanning 13 imaging modalities and 28 anatomical regions. Training open-weight LMMs with reinforcement learning using verifiable rewards improves accuracy across six medical VQA benchmarks, achieving averages of 55.85 (3B) and 58.15 (7B), with up to 77.57 on VQA-RAD and 67.76 on PathVQA, outperforming strong medical LMMs. Ablations verify that both generation and verification are necessary and that more verified data consistently helps, and a targeted contamination analysis detects no leakage from evaluation suites. By operating entirely on open literature and open-weight models, MedVLSynther offers an auditable, reproducible, and privacy-preserving path to scalable medical VQA training data.

## 1 INTRODUCTION

Large Multimodal Models (LMMs) are rapidly becoming capable assistants for biomedical discovery and clinical education, where questions must be answered by jointly interpreting medical images (e.g., X-ray, CT, microscopy) and the surrounding textual context (e.g., figure captions, narrative descriptions, etc.). Despite fast progress, training *general* medical VQA systems remains difficult because the community lacks large, openly usable, and *high-quality* training corpora.

On the evaluation side, recent benchmark (Hu et al., 2024; Ye et al., 2024) provide broad and challenging test suites, but they are designed for *assessment* rather than training and therefore offer no training splits. On the training side, existing datasets fall into three categories, each with a limitation. 1) **Manually curated** sets (Lau et al., 2018; Liu et al., 2021; He et al., 2020) are carefully annotated but are either small or bound to narrow modalities and topics, limiting coverage and transfer. 2) **Automatically generated** sets (Zhang et al., 2023b; Chen et al., 2024c) scale more easily but are typically produced by text-only LLMs that ignore visual evidence and figure–text relations, yielding noisy stems, ambiguous options, and medically dubious answers that can impede model learning. 3) **Closed, large-scale** resources (Li et al., 2024) exist but are not publicly shareable due to patient privacy, licensing, and institutional agreements, which slows open research and reproducibility. Collectively, these constraints lead to a practical bottleneck: we can *evaluate* medical VQA systems comprehensively, but we cannot *train* them broadly and transparently.

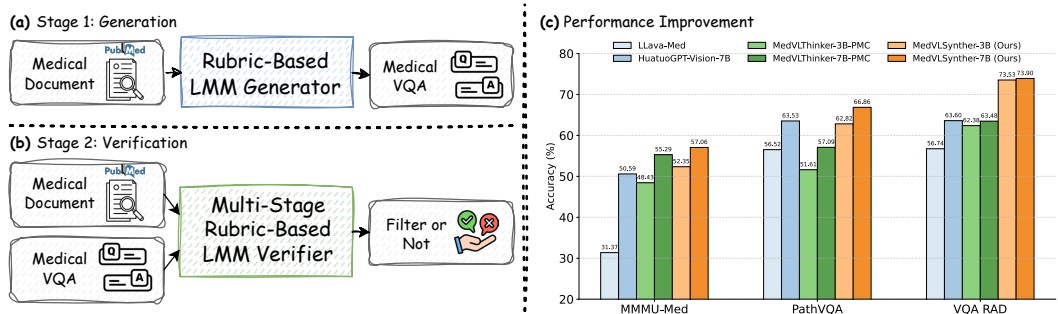

Figure 1: (a) Stage-1 generation: a rubric-guided LMM converts PubMed figures and captions into multiple-choice VQA items. (b) Stage-2 verification: a multi-stage, rubric-based LMM verifier screens items and filters low-quality ones. (c) Training open-weight students (3B/7B) on MedSyn-VQA yields consistent gains over strong medical LMM baselines.

This paper asks a simple question: *can we synthesize high-quality, auditable medical VQA data directly from open biomedical literature?* Our answer is **MedVLSynther**, a generator–verifier framework that leverages state-of-the-art open-weight LMMs (Zeng et al., 2025; Wang et al., 2025; Bai et al., 2025) to produce and automatically vet VQA triplets from figures and surrounding text in PubMed articles (Lozano et al., 2025; Roberts, 2001). The key design choice is to make both generation and verification **explicitly rubric-driven** and **context-aware**.

**Rubric-guided context-aware generation** (Figure 1 (a)). Given a figure, its caption, and the figure's in-text reference paragraph when available, the generator LMM is instructed to propose a VQA item, including question stem, multiple-choice options, and the correct answer, under a comprehensive rubric. The rubric enforces that stems are self-contained and anchored in the provided visual–textual context, that options are parallel and mutually exclusive, and that the answer can be justified from the figure and caption, not from world knowledge alone. The rubric also specifies a set of accepted *question archetypes* (e.g., recognition, localization, comparative, reasoning) and a JSON schema/format that simplifies downstream filtering and training.

**Multi-stage rubric-based verification** (Figure 1 (b)). To ensure quality, we feed the same context and the generated VQA to a verifier LMM and score it in three stages: 1) **Essential criteria** form strict pass/fail gates. Any failure discards the item. 2) **Fine-grained criteria** award positive points with justifications, and allow the verifier to surface additional criteria opportunistically. 3) **Penalty criteria** investigate common failure modes and subtract points when detected. We sum the fine-grained and penalty scores and apply a threshold to filter surviving items. This verifier is model-agnostic and can be instantiated with any open-weight LMM; in practice we find that a verifier different from the generator improves robustness.

The generator–verifier loop yields a *data pipeline* whose rules are transparent and auditable end-to-end. Because we build on open literature rather than protected clinical data, the entire pipeline, including prompts, rubric, and metadata, can be inspected and reproduced. At the same time, recent open-weight LMMs rival proprietary systems on many multimodal tasks (Zeng et al., 2025), allowing us to benefit from strong perception and reasoning while staying fully open.

The resulting medical VQA dataset, **MedSynVQA**, covering diverse modalities, subspecialties, and question archetypes. Models trained on this data with Reinforcement Learning with Verifiable Rewards (RLVR) (Guo et al., 2025; Shao et al., 2024) outperform counterparts trained on PMC-VQA (Zhang et al., 2023b), as well as the strong baseline trained on text-only medical corpora (Huang et al., 2025b). As summarized in Figure 1 (c), our training improves accuracy on MMMU-Med (Yue et al., 2024), PathVQA (He et al., 2020), and VQA-RAD (Lau et al., 2018) over strong baselines (Alshibli et al., 2025; Li et al., 2023). Meanwhile, ablations reveal that (i) both *generation* and *verification* are necessary: their synergy yields the best accuracy, and (ii) scale matters: more verified data consistently helps. We analyze topic coverage, modality distribution, and question types, and most importantly, conduct a **contamination analysis** tailored for synthetic medical VQA; we find **no** detectable leakage from the evaluation sets.

Table 1: Comparison among medical VQA datasets. MedSynVQA is open and reproducible, covering 13 modalities and 28 anatomical regions, with 13,087 questions over 14,803 images. "N/A" indicates missing statistics. "# Rate" denotes ratio of images/questions.

| Dataset | # Questions | # Images | # Rate | # Modality | # Anatomy | Annotation | Data Availability | General QA | Training Set |
|---|---|---|---|---|---|---|---|---|---|
| MedXpertQA-MM | 2,000 | 2,852 | 1.43 | 10 | 11 | Expert | Open access | Yes | No |
| GMAI-MMBench | 25,831 | 25,831 | 1.00 | 38 | N/A | Automatic | Open access | Yes | No |
| OmniMedVQA | 127,995 | 118,010 | 0.92 | 12 | 26 | Automatic | Open access | Yes | No |
| SLAKE | 14,028 | 642 | 0.05 | 3 | 5 | Expert | Open access | No | Yes |
| VQA-RAD | 3,515 | 315 | 0.09 | 3 | 3 | Expert | Open access | No | Yes |
| PathVQA | 32,799 | 4,998 | 0.15 | 2 | N/A | Automatic | Open access | No | Yes |
| PMC-VQA | 226,946 | 149,075 | 0.66 | N/A | N/A | Automatic | Open access | Yes | Yes |
| GMAI-VL-5.5M | ≈ 5,500,000 | N/A | N/A | 13 | N/A | Automatic | Not Open | Yes | Yes |
| MedSynVQA | 13,087 | 14,803 | 1.13 | 13 | 28 | Automatic | Open access | Yes | Yes |

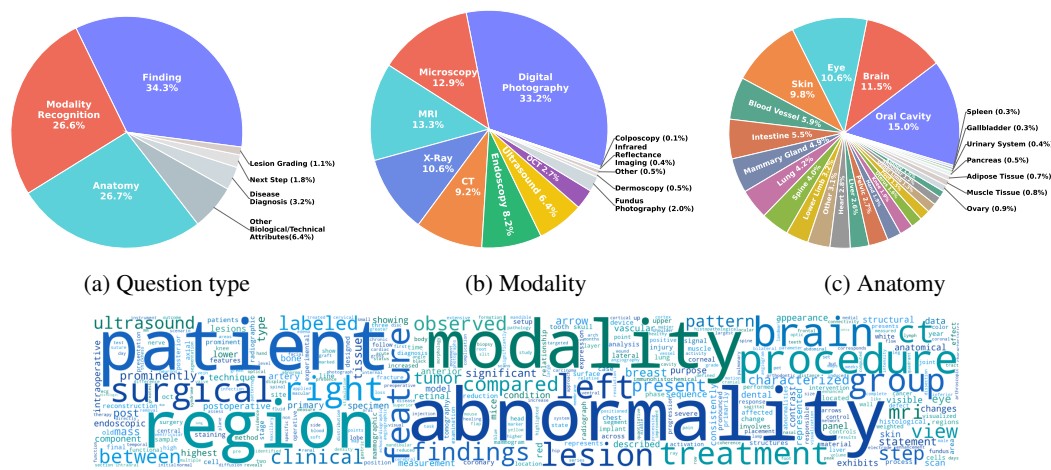

(a) Question type         (b) Modality         (c) Anatomy

(d) Word cloud for generated questions.

Figure 2: **MedSynVQA** statistics: 1) Dataset distributions for question type, imaging modality, and anatomy. 2) Word cloud for generated questions.

Our contributions are summarized as follows:

- **MedVLSynther**, a **rubric-guided, context-aware generator–verifier pipeline** that synthesizes reliable medical VQA from open biomedical articles.

- A **comprehensive rubric** for medical VQA quality, spanning essential gates, fine-grained positive criteria, and penalty criteria, together with a machine-checkable schema that supports automatic filtering and auditing.

- A **synthetic medical VQA training set** (MedSynVQA) that substantially improves medical LMMs on multiple medical VQA benchmarks and complements existing resources without relying on private patient data.

- **Transparency and reproducibility:** our pipeline operates entirely on open data and open models, enabling the community to inspect prompts, scoring rules, and filtering decisions end-to-end.

While synthetic data *cannot* replace carefully curated clinical datasets, our results indicate that *high-quality, auditable synthesis is both feasible and useful* for medical VQA. We hope **MedVLSynther** provides a practical path to scalable training data that respects privacy, encourages openness, and accelerates progress in multimodal medical intelligence.

## 2 RELATED WORKS

**Multimodal medical VQA.** Early, expert-curated datasets (Lau et al., 2018; Liu et al., 2021; He et al., 2020) established Med-VQA but remain small or modality-restricted, limiting transfer. Later,

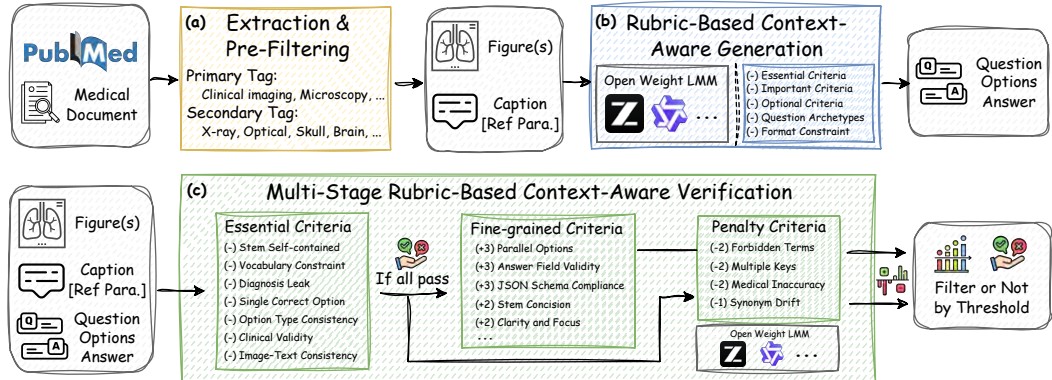

Figure 3: From PubMed documents we extract figures and reference text, then apply (a) extraction and pre-filtering by primary/secondary tags; (b) rubric-based, context-aware generation with format constraints and question archetypes; (c) multi-stage verification with essential, fine-grained, and penalty criteria. Items are retained if their rubric score exceeds a threshold.

broad benchmarks (Hu et al., 2024; Ye et al., 2024; Zuo et al., 2025) consolidated evaluation across many modalities and anatomies yet offer little or no training data, creating a supervision bottleneck. In contrast, large literature-derived corpora (Subramanian et al., 2020; Rückert et al., 2024) and especially Lozano et al. (2025)'s 24M image–caption pairs provide open, scalable raw material. Our work converts this open substrate into exam-quality VQA by coupling context-aware generation with rigorous verification, bridging the gap between expansive evaluation suites and accessible training data.

**Synthetic data generation for multimodal medical VQA.** Prior synthetic pipelines scale supervision but suffer quality issues: Li et al. (2023)'s self-instruct approach and Zhang et al. (2023b)'s 227k auto-generated pairs (largely from text-only LLMs) can omit modality cues, produce ambiguous stems, and yield visually ungrounded answers; broader compilations like Li et al. (2024) are closed, while modality-specific (Hu et al.) sets remain narrow. These limitations motivate a quality-first strategy: we condition on figures, captions, and in-text references and enforce a rubric-guided generator plus a multi-stage verifier to filter low-quality items, yielding reliable, open data suitable for training medical LMMs without relying on private images.

**Multimodal models, medical adaptation, and reasoning.** General LVLMs (Hurst et al., 2024; Comanici et al., 2025; Wang et al., 2025; Bai et al., 2025; Liu et al., 2024; An et al., 2025; An et al.) acquire instruction following via visual SFT, while medical variants (Tu et al., 2024; Luo et al., 2023; Alshibli et al., 2025; Liu et al., 2023; Wu et al., 2025; Chen et al., 2024b; Zhou et al., 2024; Wu et al., 2023) add in-domain pretraining and SFT/RL for clinical competence. Recent deliberate-reasoning models (Jaech et al., 2024) show that reinforcement learning with verifiable rewards (e.g., GRPO (Guo et al., 2025)) can surpass SFT-only methods on multi-step problems, and early medical efforts point the same way but lack open, high-quality multimodal supervision. Our rubric-verified VQA corpus supplies that missing signal and pairs naturally work for RLVR, contributing auditable and trustworthy visual reasoning in open-weight medical LMMs (Chen et al., 2024c; Su et al., 2025).

## 3 MEDVLSYNTHER AND MEDSYNVQA

Our goal is to synthesize high-quality, clinically valid multiple-choice VQA (MC-VQA) examples directly from biomedical papers (Lozano et al., 2025). We cast the task as a **Generator–Verifier** pipeline driven by Large Multimodal Models (LMMs): a *rubric-guided generator* produces MC-VQA items from figures and text, and a *multi-stage rubric-guided verifier* performs automatic quality control before data are admitted to the final corpus (Figure 3).

## 3.1 DATA SOURCE, EXTRACTION, AND PRE-FILTERING

**Source.** We build on Biomedica (Lozano et al., 2025), a large-scale extraction of figures and figure-level metadata from the PubMed Central Open-Access (PMC-OA) collection (Roberts, 2001). For each paper we ingest: 1) The figure image(s) (a single caption may reference up to 6 images), 2) The figure caption. 3) The corresponding *figure references* in the main text (when present).

Samples missing either images or a caption are discarded.

**Pre-filtering.** We retain items annotated by Lozano et al. (2025) with the **primary labels**: *Clinical imaging* and *Microscopy*, and 25 **secondary subtypes** (e.g., *x-ray radiography*, *optical coherence tomography*, *skull*, *brain*, etc.). After pre-filtering we obtain **23,788** figure-caption(-reference) triplets.

We denote each pre-filtered sample by

$$x = (\mathcal{I}, \mathcal{C}, \mathcal{R}), \tag{1}$$

where $\mathcal{I}$ is one or more images, $\mathcal{C}$ the caption, and $\mathcal{R}$ the in-text references.

**Choice of generator and verifier LMMs.** We use state-of-the-art *open-weight LMM* capable of long-context vision-language reasoning: GLM-4.5V-108B (Zeng et al., 2025), InterVL-3.5-38B (Wang et al., 2025), and Qwen2.5-VL-72B (Bai et al., 2025). Unless otherwise noted, GLM-4.5V-108B serves as the default generator due to its strong instruction-following and image-grounding performance. The rubric and strict JSON schema make the output predictable and machine-verifiable.

## 3.2 RUBRIC-BASED, CONTEXT-AWARE VQA GENERATION

Given $x$, the *generator LMM* $G_\theta$ produces a *5-option* MC-VQA instance in strict JSON format: $y = \{q, \text{options}\{A..E\}, \text{answer} \in \{A..E\}\}$ . Generation is *context-aware* the model receives the image(s) together with $\mathcal{C}$ and $\mathcal{R}$. To ensure exam-quality items, the prompt instills the role of an *expert medical-education item writer* and enforces a *self-check rubric*.

- **Essential (must pass before output):** 1) *Stem self-contained* (no "caption/context" mentions); 2) *Image–content alignment* (requires inspecting specific visual features); 3) *Implicit use of caption facts without answer leakage*; 4) *Exactly one best answer*; 5) *Medical correctness* (modality, anatomy, terminology).

- **Important (strongly recommended):** cognitive level is over application; strong, parallel distractors; clear focus on a single concept.

- **Optional:** localization or quantitative details when clearly supported.

A small set of *question archetypes* (*i.e.*, Finding/Abnormality Identification, Modality Recognition, Anatomy/Localization, Other Biological/Technical Attributes, Disease Diagnosis, Next Step, and Lesion Grading) reduces prompt entropy and encourages clinically meaningful questions.

## 3.3 MULTI-STAGE, RUBRIC-BASED, CONTEXT-AWARE VERIFICATION

While the generator is reliable, automatic verification is essential for scale and precision. Given $x$ and a candidate MC-VQA $y$, the *verifier LMM* $V_\phi$ is prompted to operate in two roles, *Referee* and *Critic*, and to return only a structured rubric with binary scores. Verification is also *context-aware*: $V_\phi$ sees the same images, caption, and references as $G_\theta$ plus the proposed MC-VQA.

**Stage-1: Essential screening (hard gate).** The Referee evaluates seven non-negotiable items; a sample *must pass all* to proceed: 1) Stem Self-contained; 2) Vocabulary Constraint (no unsupported clinical facts); 3) Diagnosis Leak (no verbatim restatement from sources); 4) Single Correct Option; 5) Option Type Consistency (same semantic type); 6) Clinical Validity (terminology/modality/anatomy); 7) Image–Text Consistency.

Items are scored $\{0, 5\}$ with a fair, rule-based mindset. During this stage, we remove instances that the verifier cannot grade (e.g., malformed JSON), leaving **23,635** candidates. After applying the essential filter, **22,903** remain.

**Stage-2: Fine-grained positive criteria (bonus points).** The Critic now assumes *the item is not excellent* and awards points *only on irrefutable evidence* We query 4–8 bonus criteria (binary, with weights *Important* $= 3$ or $4$, *Optional* $= 1$ or $2$), including: 1) Plausible Distractors (every distractor is a strong near-miss); 2) Parallel Options (length/structure uniformity); 3) Stem Concision (less than two sentences and concise); 4) Clarity and Focus (single, unambiguous question); 5) Answer-field Validity (answer exists, matches an option); 6) JSON Schema Compliance (exact keys, no extras).

The Critic denies a criterion if it can imagine a slightly better wording or distractor, pushing precision over recall.

**Stage-3: Penalty criteria (error hunting).** Finally, the Critic actively searches for pitfalls (negative weights): 1) Forbidden Terms ($-2$; stem contains "caption/context"); 2) Synonym Drift ($-1$; introduces unsupported specific facts); 3) Multiple Keys ($-2$), and Medical Inaccuracy ($-2$).

Each pitfall is triggered only with a concrete reason.

## 3.4 Aggregation and acceptance rule

Let $\mathcal{P}$ be the set of positive (Important $\cup$ Optional) criteria with weights $w_i > 0$ and binary scores $s_i \in \{0, w_i\}$. Let $\mathcal{N}$ be the pitfalls with $w_j < 0$ and scores $p_j \in \{0, w_j\}$. We compute a *normalized quality score*:

$$S(x,y) = \text{clip}_{[0,1]}\left(\frac{\sum_{i \in \mathcal{P}} s_i + \sum_{j \in \mathcal{N}} p_j}{\sum_{i \in \mathcal{P}} w_i}\right). \tag{2}$$

Candidates passing Stage-1 are *accepted* if $S(x,y) \geq \tau$ with $\tau = 0.9670$. This high threshold emphasizes precision while keeping a useful yield; it results in **13,087** MC-VQA items, which we call **MedSynVQA**.

## 3.5 Training medical LMMs with MedSynVQA

We use our synthesized corpus to train medical LMMs with two LMM finetuning approaches.

**Supervised Fine-Tuning (SFT).** Following MedVLThinker, we elicit *thinking traces* with GLM-4.5V-108B and perform SFT on (thinking trace, answer) pairs. The supervision emphasizes clinically grounded reasoning paths while preserving the strict answer format.

**RL with Verbal Rewards (RLVR).** We then apply GRPO on *answers only* (no trace optimization), again mirroring hyper-parameters from Huang et al. (2025b). The reward promotes exact-match accuracy and adherence to the schema without over-fitting to any single imaging modality.

## 4 Experiments

### 4.1 Setup

**Models.** Unless otherwise stated, we finetune two open-weight LMMs Qwen2.5-VL 3B and 7B Instruct (Bai et al., 2025), using the same training schedule, image resolution, tokenization, and optimization hyper-parameters as Huang et al. (2025b). We use our rubric-guided generator–verifier pipeline to synthesize training items from PubMed figures and captions (Figure 3), and we train students either with SFT or RLVR. Unless otherwise noted, experiments use **5K** samples.

**Benchmarks and metric.** We follow Huang et al. (2025b) evaluation suite and scripts, reporting multiple-choice accuracy on six medical VQA benchmarks: MMMU medical split (MMMU-Med) (Yue et al., 2024), (MedX-M) (Zuo et al., 2025), PathVQA (He et al., 2020), PMC-VQA (Zhang et al., 2023b), SLAKE (Liu et al., 2021), and VQA-RAD (Lau et al., 2018).

Table 2: Generator–Verifier pipeline ablation. Rubric-guided generation outperforms PMC-Style Text-Image Generation, and adding verification yields the best average accuracy. Cells are shaded by accuracy; darker is better.

| Model | MMMU | MedX-M | PathVQA | PMC | SLAKE | VQA-RAD | Avg. |
|---|---|---|---|---|---|---|---|
| Qwen2.5-VL-3B-Instruct | 44.12 | 20.69 | 61.96 | 44.77 | 61.30 | 62.01 | 49.14 |
| PMC-Style Text-only Generation | 46.47 | 20.8 | 62.43 | 51.03 | 73.08 | 71.69 | 54.25 |
| PMC-Style Text-Image Generation | 48.82 | 20.40 | 63.38 | 51.08 | 73.08 | 72.06 | 54.80 |
| Rubric Context-Aware Generation | 52.35 | 20.60 | 62.49 | 51.83 | 70.43 | 70.59 | 54.72 |
| + Rubric Context-Aware Verification | 52.35 | 21.40 | 62.82 | 50.23 | 74.76 | 73.53 | 55.85 |
| Qwen2.5-VL-7B-Instruct | 52.94 | 18.89 | 65.39 | 49.30 | 65.71 | 68.75 | 53.50 |
| PMC-Style Text-only Generation | 54.71 | 22.15 | 63.89 | 53.23 | 67.07 | 65.44 | 54.41 |
| PMC-Style Text-Image Generation | 51.76 | 21.70 | 64.31 | 53.43 | 68.03 | 71.69 | 55.15 |
| Rubric Context-Aware Generation | 58.24 | 23.50 | 65.41 | 53.83 | 68.03 | 75.00 | 57.33 |
| + Rubric Context-Aware Verification | 57.06 | 23.15 | 66.36 | 53.78 | 67.79 | 77.21 | 57.56 |

Table 3: Dataset scale ablation. Effect of the number of MedSynVQA training items (1k–13k) on downstream accuracy. Performance improves with scale, with diminishing returns beyond 5k examples. "N/A" denotes zero-shot (no additional training). Cells are shaded by accuracy.

| Model | Scale | MMMU | MedX-M | PathVQA | PMC | SLAKE | VQA-Rad | Avg. |
|---|---|---|---|---|---|---|---|---|
| Qwen2.5-VL 3B-Instruct | N/A | 44.12 | 20.69 | 61.96 | 44.77 | 61.30 | 62.01 | 49.14 |
| | 1000 | 50.59 | 20.20 | 63.18 | 48.37 | 65.87 | 67.65 | 52.64 |
| | 2000 | 47.06 | 19.95 | 64.07 | 47.27 | 74.04 | 76.84 | 54.87 |
| | 5000 | 52.35 | 21.40 | 62.82 | 50.23 | 74.76 | 73.53 | 55.85 |
| | 10000 | 48.82 | 20.55 | 63.44 | 49.87 | 72.84 | 74.63 | 55.03 |
| | Full | 51.76 | 22.30 | 63.03 | 48.92 | 72.60 | 72.43 | 55.17 |
| Qwen2.5-VL 7B-Instruct | N/A | 52.94 | 18.89 | 65.39 | 49.30 | 65.71 | 68.75 | 53.50 |
| | 1000 | 57.65 | 21.60 | 65.53 | 50.93 | 68.27 | 65.81 | 54.96 |
| | 2000 | 60.00 | 22.35 | 67.76 | 51.18 | 67.31 | 73.16 | 56.96 |
| | 5000 | 57.06 | 23.15 | 66.36 | 53.78 | 67.79 | 77.21 | 57.56 |
| | 10000 | 57.06 | 22.45 | 66.86 | 52.73 | 71.88 | 73.90 | 57.48 |
| | Full | 55.88 | 22.10 | 65.56 | 55.43 | 72.36 | 77.57 | 58.15 |

**Baselines.** We compare against strong general-purpose and medical LMMs used in Huang et al. (2025b), including Gemma3 4B (Team et al., 2025), Qwen2.5-VL-3B/7B-Instruct, MedGemma 4B (Sellergren et al., 2025), LLaVA-Med (Li et al., 2023), HuatouGPT-Vision-7B (Chen et al., 2024c), and MedVLThinker (Huang et al., 2025b), strong baselines trained solely on text-only data.

## 4.2 RESULTS

**Ablation on the Generator–Verifier pipeline.** Table 2 studies each stage of our pipeline. We begin from zero-shot Qwen2.5-VL students and add 1) PMC-style text-only question generation, 2) rubric-guided context-aware generation, and 3) rubric-aware verification. For 3B student, the base model averages 49.14. Text-only generation lifts the average to 54.80. Switching to rubric-guided, context-aware generation performs similarly on average (54.72). Adding verification yields the best average, 55.85, with large gains on clinically grounded datasets. For 7B student, the base model averages 53.50. Text-only generation yields 55.15, rubric-guided generation 57.33, and with verification we obtain 57.56 and again improving across benchmarks. Overall, rubric guidance already outperforms a PMC-style text-only recipe, and multi-stage verification supplies the remaining headroom, producing the best average in both scales (Table 2). The trend aligns with the high-level improvements visualized in Figure 1.

**How much synthesized data do we need?** We vary the number of MedSynVQA training items from 1K to 13K (Table 3). For 3B student. Accuracy increases from 52.64 (1K) to 55.85 (5K), then plateaus at 55.03 (10K) and 55.17 (Full). For 7B student. The curve is similar: 54.96 (1K), 56.96 (2K), 57.56 (5K) with a slight dip to 57.48 at 10K, and a peak of 58.15 with the full dataset. This tendency suggests the potential for further refinement of the filtering method. Moreover, to reduce computational cost, we use 5K items as the default scale in subsequent experiments.

Table 4: Choice of generator and verifier LMMs. We vary the generator and verifier. Higher-capacity generator/verifier pairs produce higher-quality data and consistently improve the final average accuracy. "N/A" indicates the zero-shot performance. Cells are shaded by accuracy.

| Model | Generator | Verifier | MMMU | MedX-M | PathVQA | PMC | SLAKE | VQA-Rad | Avg. |
|---|---|---|---|---|---|---|---|---|---|
| | N/A | N/A | 44.12 | 20.69 | 61.96 | 44.77 | 61.30 | 62.01 | 49.14 |
| Qwen2.5-VL | GLM-4.5V 108B | Qwen2.5-VL 72B | 52.35 | 21.40 | 62.82 | 50.23 | 74.76 | 73.53 | 55.85 |
| 3B-Instruct | GLM-4.5V 108B | GLM-4.5V 108B | 51.18 | 20.30 | 63.56 | 50.63 | 71.63 | 70.22 | 54.59 |
| | Qwen2.5-VL 72B | GLM-4.5V 108B | 47.65 | 21.50 | 62.37 | 48.87 | 73.32 | 69.85 | 53.93 |
| | InternVL3.5 38B | GLM-4.5V 108B | 49.41 | 21.90 | 61.81 | 51.98 | 74.76 | 71.32 | 55.20 |
| | N/A | N/A | 52.94 | 18.89 | 65.39 | 49.30 | 65.71 | 68.75 | 53.50 |
| Qwen2.5-VL | GLM-4.5V 108B | Qwen2.5-VL 72B | 57.06 | 23.15 | 66.36 | 53.78 | 67.79 | 77.21 | 57.56 |
| 7B-Instruct | GLM-4.5V 108B | GLM-4.5V 108B | 58.82 | 23.65 | 67.22 | 54.48 | 71.15 | 73.16 | 58.08 |
| | Qwen2.5-VL 72B | GLM-4.5V 108B | 56.47 | 22.55 | 67.25 | 52.38 | 67.07 | 72.79 | 56.42 |
| | InternVL3.5 38B | GLM-4.5V 108B | 57.65 | 23.30 | 66.12 | 53.58 | 70.67 | 75.37 | 57.78 |

Table 5: Training approach and data source ablation. Comparing SFT and RLVR using three sources: PMC (image-text), m23k (text-only), MedSynVQA. RL consistently outperforms SFT, and MedSynVQA leads to the highest averages across all benchmarks. Cells are shaded by accuracy.

| Model | MMMU | MedX-M | PathVQA | PMC | SLAKE | VQA-RAD | Avg. |
|---|---|---|---|---|---|---|---|
| Qwen2.5-VL-3B-Instruct | 44.12 | 20.69 | 61.96 | 44.77 | 61.30 | 62.01 | 49.14 |
| SFT (PMC) | 47.84 | 21.46 | 52.76 | 54.55 | 65.79 | 58.58 | 50.16 |
| SFT (m23k) | 32.55 | 16.00 | 42.74 | 28.53 | 43.91 | 33.09 | 32.80 |
| SFT (MedSynVQA) | 48.82 | 20.90 | 63.12 | 47.57 | 54.33 | 59.93 | 49.11 |
| RL (PMC) | 48.43 | 21.51 | 51.61 | 54.22 | 75.56 | 62.38 | 52.28 |
| RL (m23k) | 52.16 | 22.90 | 62.28 | 47.32 | 63.38 | 71.08 | 53.19 |
| RL (MedSynVQA) | 52.35 | 21.40 | 62.82 | 50.23 | 74.76 | 73.53 | 55.85 |
| Qwen2.5-VL-7B-Instruct | 52.94 | 18.89 | 65.39 | 49.30 | 65.71 | 68.75 | 53.50 |
| SFT (PMC) | 49.80 | 21.39 | 53.02 | 54.67 | 67.71 | 57.72 | 50.72 |
| SFT (m23k) | 46.86 | 16.40 | 56.35 | 34.58 | 54.97 | 53.80 | 43.83 |
| SFT (MedSynVQA) | 49.41 | 20.90 | 64.81 | 50.08 | 59.62 | 66.54 | 51.89 |
| RL (PMC) | 55.29 | 24.11 | 57.09 | 55.38 | 66.59 | 63.48 | 53.66 |
| RL (m23k) | 56.86 | 24.43 | 66.83 | 50.67 | 65.79 | 64.71 | 54.88 |
| RL (MedSynVQA) | 57.06 | 23.15 | 66.36 | 53.78 | 67.79 | 77.21 | 57.56 |

**Which generator and verifier LMMs should we use?** We next vary the capacity and identity of the generator and verifier LMMs used during data synthesis (Table 4). For the 3B student, pairing a GLM-4.5V-108B generator with a Qwen2.5-VL-72B verifier yields the best average 55.85; other high-capacity pairs are close. For the 7B student, the same open-weight verifier gives 57.56 with a GLM-108B generator, while using GLM-108B as both generator and verifier further nudges the average to 58.08. We keep the Qwen2.5-VL-72B verifier for the main results to maximize reproducibility with open weights, but Table 4 indicates that stronger verifier capacity translates to higher downstream accuracy.

**Training approach and data source ablation.** Table 5 compares SFT vs RL from verification reward (RL) across three data sources: PMC-VQA (image–text pairs) (Zhang et al., 2023b), m23k (text-only) (Huang et al., 2025a), and MedSynVQA. 1) RL outperform SFT for both 3B and 7B models, across all data source. 2) Under RL the MedSynVQA signal is the strongest, giving the best average on both 3B (55.85) and 7B (57.56). The results indicate that rubric-based context-aware MedSynVQA dataset are more effective training source than the previous synthetic PMC-VQA (Zhang et al., 2023b) and the text-only one (Huang et al., 2025b).

**Comparisons.** Table 6 summarizes head-to-head results on the full benchmark suite. Our students trained with MedSynVQA achieve 55.85 (3B) and 58.15 (7B), state-of-the-art averages among open-weight models considered. Notably, our *3B* student surpasses MedVLThinker-*7B* by +0.97 and all other *3–7B* baselines; the *7B* student improves over the best prior MedVLThinker-7B by +3.27. Gains are consistent across datasets, with strong results on VQA-RAD (up to 77.57).

Table 6: Comparison to baselines. Average and per-benchmark accuracy of general-purpose and medical LMMs versus models trained with MedSynVQA. Both MedVLSynther 3B and 7B achieve the best average across benchmarks, demonstrating strong gains at small and medium scales.

| Model | MMMU | MedX-M | PathVQA | PMC | SLAKE | VQA-Rad | Avg. |
|---|---|---|---|---|---|---|---|
| General LLM | | | | | | | |
| Gemme 3 4B | 46.67 | 21.89 | 59.24 | 44.42 | 66.59 | 56.86 | 49.28 |
| Qwen2.5-VL-3B-Instruct | 44.12 | 20.69 | 61.96 | 44.77 | 61.30 | 62.01 | 49.14 |
| Qwen2.5-VL-7B-Instruct | 52.94 | 18.89 | 65.39 | 49.30 | 65.71 | 68.75 | 53.50 |
| Medical LLM | | | | | | | |
| MedGemma 4B | 32.55 | 8.17 | 59.64 | 42.73 | 83.49 | 78.55 | 50.86 |
| MedGemma 27B | 35.88 | 12.13 | 62.09 | 36.75 | 77.40 | 72.67 | 49.49 |
| Llava Med V1.5 7B | 31.37 | 22.56 | 56.52 | 34.28 | 62.82 | 56.74 | 44.05 |
| HuatuoGPT-Vision-7B | 50.59 | 22.00 | 63.53 | 53.39 | 75.00 | 63.60 | 54.69 |
| MedVLThinker-3B | 52.16 | 22.90 | 62.28 | 47.32 | 63.38 | 71.08 | 53.19 |
| MedVLThinker-7B | 56.86 | 24.43 | 66.83 | 50.67 | 65.79 | 64.71 | 54.88 |
| MedVLSynther-3B | 52.35 | 21.40 | 62.82 | 50.23 | 74.76 | 73.53 | 55.85 |
| MedVLSynther-7B | 55.88 | 22.10 | 65.56 | 55.43 | 72.36 | 77.57 | 58.15 |

**Case study.** Figure 4 presents two cases, revealing deep comprehension with context for our generator and the leakage rejection by our verifier. Please refer to the **appendix** for more details.

**Contamination analysis.** We practice contamination analysis between MedSynVQA and the evaluation suites in the **appendix**. **No** overlaps were found under this protocol.

## 5 CONCLUSIONS

MedVLSynther shows that high-quality, auditable medical VQA data can be synthesized at scale from open biomedical literature by pairing rubric-guided, context-aware generation with a multi-stage verifier. The resulting MedSynVQA delivers consistent gains for open-weight LMMs across six benchmarks and ablations confirm that both the generator and verifier are necessary. Operating entirely on open data and models, the approach offers a reproducible, privacy-preserving, and transparent path to supervision for medical VQA.

## ACKNOWLEDGMENTS

This work was partially funded by an unrestricted gift from Google.

## REFERENCES

Ahmad Alshibli, Yakoub Bazi, Mohamad Mahmoud Al Rahhal, and Mansour Zuair. Vision-biollm: Large vision language model for visual dialogue in biomedical imagery. *Biomedical Signal Processing and Control*, 103:107437, 2025.

Zhaochong An, Guolei Sun, Yun Liu, Runjia Li, Min Wu, Ming-Ming Cheng, Ender Konukoglu, and Serge Belongie. Multimodality helps few-shot 3d point cloud semantic segmentation. In *The Thirteenth International Conference on Learning Representations*.

Zhaochong An, Guolei Sun, Yun Liu, Runjia Li, Junlin Han, Ender Konukoglu, and Serge Belongie. Generalized few-shot 3d point cloud segmentation with vision-language model. In *Proceedings of the Computer Vision and Pattern Recognition Conference*, pp. 16997–17007, 2025.

Shuai Bai, Keqin Chen, Xuejing Liu, Jialin Wang, Wenbin Ge, Sibo Song, Kai Dang, Peng Wang, Shijie Wang, Jun Tang, et al. Qwen2. 5-vl technical report. *arXiv preprint arXiv:2502.13923*, 2025.

---

**Case 1: Context-Aware Generation**

**Question**: Which of the following best describes the vertebral anomaly shown in this image?
**Options**: (A) Six lumbar vertebrae with incomplete sacral integration (B) Five lumbar vertebrae with complete sacral fusion ...
**Answer**: (A)

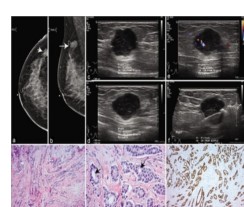

---

**Caption**: Lumbar transitional vertebrae
**Context**: The subject was diagnosed as having CLBP and a lumbar transitional vertebra. Computer tomography (CT) showed six lumbar vertebrae, which is one more lumbar vertebra than a normal person ... This indicates the first sacrum is not completely integrated...
**Pass Verification**: **True**

---

**Case 2: Leakage Rejection by Verifier**

**Question**: A 37-year-old woman presents with a palpable mass in the upper outer quadrant of the right breast. Imaging and biopsy reveal a circumscribed hypoechoic mass with flow, invasive ductal carcinoma, grade 1, with predominant tubular formation, mild nuclear atypia, and strong diffuse ER positivity. Which histopathological feature is most consistent with the tumor grade?
**Options**: ... (C) Well-differentiated with predominant tubular formation and mild nuclear atypia ...
**Answer**: (C)

---

**Caption**: ...invasive ductal carcinoma, grade 1, ...showing predominant tubular formation (arrows), mild nuclear atypia ... ER immunostain showing strong diffuse positivity ...
**Context**: ...
**Pass Verification**: **False**

Figure 4: Examples of context-aware generation and leakage rejection by the verifier.

Jianlv Chen, Shitao Xiao, Peitian Zhang, Kun Luo, Defu Lian, and Zheng Liu. Bge m3-embedding: Multi-lingual, multi-functionality, multi-granularity text embeddings through self-knowledge distillation. *arXiv preprint arXiv:2402.03216*, 2024a.

Junying Chen, Zhenyang Cai, Ke Ji, Xidong Wang, Wanlong Liu, Rongsheng Wang, Jianye Hou, and Benyou Wang. Huatuogpt-o1, towards medical complex reasoning with llms. *arXiv preprint arXiv:2412.18925*, 2024b.

Junying Chen, Chi Gui, Ruyi Ouyang, Anningzhe Gao, Shunian Chen, Guiming Hardy Chen, Xidong Wang, Ruifei Zhang, Zhenyang Cai, Ke Ji, et al. Huatuogpt-vision, towards injecting medical visual knowledge into multimodal llms at scale. *arXiv preprint arXiv:2406.19280*, 2024c.

Gheorghe Comanici, Eric Bieber, Mike Schaekermann, Ice Pasupat, Noveen Sachdeva, Inderjit Dhillon, Marcel Blistein, Ori Ram, Dan Zhang, Evan Rosen, et al. Gemini 2.5: Pushing the frontier with advanced reasoning, multimodality, long context, and next generation agentic capabilities. *arXiv preprint arXiv:2507.06261*, 2025.

Matthijs Douze, Alexandr Guzhva, Chengqi Deng, Jeff Johnson, Gergely Szilvasy, Pierre-Emmanuel Mazaré, Maria Lomeli, Lucas Hosseini, and Hervé Jégou. The faiss library. *arXiv preprint arXiv:2401.08281*, 2024.

Daya Guo, Dejian Yang, Haowei Zhang, Junxiao Song, Ruoyu Zhang, Runxin Xu, Qihao Zhu, Shirong Ma, Peiyi Wang, Xiao Bi, et al. Deepseek-r1: Incentivizing reasoning capability in llms via reinforcement learning. *arXiv preprint arXiv:2501.12948*, 2025.

Xuehai He, Yichen Zhang, Luntian Mou, Eric Xing, and Pengtao Xie. Pathvqa: 30000+ questions for medical visual question answering. *arXiv preprint arXiv:2003.10286*, 2020.

Xinyue Hu et al. Medical-cxr-vqa dataset: A large-scale llm-enhanced medical dataset for visual question answering on chest x-ray images.

Yutao Hu, Tianbin Li, Quanfeng Lu, Wenqi Shao, Junjun He, Yu Qiao, and Ping Luo. Omnimedvqa: A new large-scale comprehensive evaluation benchmark for medical lvlm. In *Proceedings of the IEEE/CVF Conference on Computer Vision and Pattern Recognition*, pp. 22170–22183, 2024.

Xiaoke Huang, Juncheng Wu, Hui Liu, Xianfeng Tang, and Yuyin Zhou. m1: Unleash the potential of test-time scaling for medical reasoning with large language models. *arXiv preprint arXiv:2504.00869*, 2025a.

Xiaoke Huang, Juncheng Wu, Hui Liu, Xianfeng Tang, and Yuyin Zhou. Medvlthinker: Simple baselines for multimodal medical reasoning. *arXiv preprint arXiv:2508.02669*, 2025b.

Aaron Hurst, Adam Lerer, Adam P Goucher, Adam Perelman, Aditya Ramesh, Aidan Clark, AJ Ostrow, Akila Welihinda, Alan Hayes, Alec Radford, et al. Gpt-4o system card. *arXiv preprint arXiv:2410.21276*, 2024.

Gabriel Ilharco, Mitchell Wortsman, Nicholas Carlini, Rohan Taori, Achal Dave, Vaishaal Shankar, Hongseok Namkoong, John Miller, Hannaneh Hajishirzi, Ali Farhadi, et al. Openclip. *Zenodo*, 2021.

Aaron Jaech, Adam Kalai, Adam Lerer, Adam Richardson, Ahmed El-Kishky, Aiden Low, Alec Helyar, Aleksander Madry, Alex Beutel, Alex Carney, et al. Openai o1 system card. *arXiv preprint arXiv:2412.16720*, 2024.

Jason J Lau, Soumya Gayen, Asma Ben Abacha, and Dina Demner-Fushman. A dataset of clinically generated visual questions and answers about radiology images. *Scientific data*, 5(1):1–10, 2018.

Chunyuan Li, Cliff Wong, Sheng Zhang, Naoto Usuyama, Haotian Liu, Jianwei Yang, Tristan Naumann, Hoifung Poon, and Jianfeng Gao. Llava-med: Training a large language-and-vision assistant for biomedicine in one day. *Advances in Neural Information Processing Systems*, 36: 28541–28564, 2023.

Tianbin Li, Yanzhou Su, Wei Li, Bin Fu, Zhe Chen, Ziyan Huang, Guoan Wang, Chenglong Ma, Ying Chen, Ming Hu, et al. Gmai-vl & gmai-vl-5.5 m: A large vision-language model and a comprehensive multimodal dataset towards general medical ai. *arXiv preprint arXiv:2411.14522*, 2024.

Bo Liu, Li-Ming Zhan, Li Xu, Lin Ma, Yan Yang, and Xiao-Ming Wu. Slake: A semantically-labeled knowledge-enhanced dataset for medical visual question answering. In *2021 IEEE 18th international symposium on biomedical imaging (ISBI)*, pp. 1650–1654. IEEE, 2021.

Fenglin Liu, Tingting Zhu, Xian Wu, Bang Yang, Chenyu You, Chenyang Wang, Lei Lu, Zhangdaihong Liu, Yefeng Zheng, Xu Sun, et al. A medical multimodal large language model for future pandemics. *NPJ Digital Medicine*, 6(1):226, 2023.

Haotian Liu, Chunyuan Li, Yuheng Li, and Yong Jae Lee. Improved baselines with visual instruction tuning. In *Proceedings of the IEEE/CVF conference on computer vision and pattern recognition*, pp. 26296–26306, 2024.

Alejandro Lozano, Min Woo Sun, James Burgess, Liangyu Chen, Jeffrey J Nirschl, Jeffrey Gu, Ivan Lopez, Josiah Aklilu, Anita Rau, Austin Wolfgang Katzer, et al. Biomedica: An open biomedical image-caption archive, dataset, and vision-language models derived from scientific literature. In *Proceedings of the Computer Vision and Pattern Recognition Conference*, pp. 19724–19735, 2025.

Yizhen Luo, Jiahuan Zhang, Siqi Fan, Kai Yang, Yushuai Wu, Mu Qiao, and Zaiqing Nie. Biomedgpt: Open multimodal generative pre-trained transformer for biomedicine. *arXiv preprint arXiv:2308.09442*, 2023.

Ankit Pal, Logesh Kumar Umapathi, and Malaikannan Sankarasubbu. Medmcqa: A large-scale multi-subject multi-choice dataset for medical domain question answering. In *Conference on health, inference, and learning*, pp. 248–260. PMLR, 2022.

Richard J Roberts. Pubmed central: The genbank of the published literature, 2001.

Johannes Rückert, Louise Bloch, Raphael Brüngel, Ahmad Idrissi-Yaghir, Henning Schäfer, Cynthia S Schmidt, Sven Koitka, Obioma Pelka, Asma Ben Abacha, Alba G. Seco de Herrera, et al. Rocov2: Radiology objects in context version 2, an updated multimodal image dataset. *Scientific Data*, 11(1):688, 2024.

Andrew Sellergren, Sahar Kazemzadeh, Tiam Jaroensri, Atilla Kiraly, Madeleine Traverse, Timo Kohlberger, Shawn Xu, Fayaz Jamil, Cían Hughes, Charles Lau, et al. Medgemma technical report. *arXiv preprint arXiv:2507.05201*, 2025.

Zhihong Shao, Peiyi Wang, Qihao Zhu, Runxin Xu, Junxiao Song, Xiao Bi, Haowei Zhang, Mingchuan Zhang, YK Li, Yang Wu, et al. Deepseekmath: Pushing the limits of mathematical reasoning in open language models. *arXiv preprint arXiv:2402.03300*, 2024.

Yanzhou Su, Tianbin Li, Jiyao Liu, Chenglong Ma, Junzhi Ning, Cheng Tang, Sibo Ju, Jin Ye, Pengcheng Chen, Ming Hu, et al. Gmai-vl-r1: Harnessing reinforcement learning for multimodal medical reasoning. *arXiv preprint arXiv:2504.01886*, 2025.

Sanjay Subramanian, Lucy Lu Wang, Sachin Mehta, Ben Bogin, Madeleine Van Zuylen, Sravanthi Parasa, Sameer Singh, Matt Gardner, and Hannaneh Hajishirzi. Medicat: A dataset of medical images, captions, and textual references. *arXiv preprint arXiv:2010.06000*, 2020.

Gemma Team, Aishwarya Kamath, Johan Ferret, Shreya Pathak, Nino Vieillard, Ramona Merhej, Sarah Perrin, Tatiana Matejovicova, Alexandre Ramé, Morgane Rivière, et al. Gemma 3 technical report. *arXiv preprint arXiv:2503.19786*, 2025.

Tao Tu, Shekoofeh Azizi, Danny Driess, Mike Schaekermann, Mohamed Amin, Pi-Chuan Chang, Andrew Carroll, Charles Lau, Ryutaro Tanno, Ira Ktena, et al. Towards generalist biomedical ai. *Nejm Ai*, 1(3):AIoa2300138, 2024.

Siddharth Tumre, Sangameshwar Patil, and Alok Kumar. Improved near-duplicate detection for aggregated and paywalled news-feeds. In *Proceedings of the 2025 Conference of the Nations of the Americas Chapter of the Association for Computational Linguistics: Human Language Technologies (Volume 3: Industry Track)*, pp. 979–987, 2025.

Weiyun Wang, Zhangwei Gao, Lixin Gu, Hengjun Pu, Long Cui, Xingguang Wei, Zhaoyang Liu, Linglin Jing, Shenglong Ye, Jie Shao, et al. Internvl3. 5: Advancing open-source multimodal models in versatility, reasoning, and efficiency. *arXiv preprint arXiv:2508.18265*, 2025.

Chaoyi Wu, Xiaoman Zhang, Ya Zhang, Yanfeng Wang, and Weidi Xie. Towards generalist foundation model for radiology by leveraging web-scale 2d&3d medical data. *arXiv preprint arXiv:2308.02463*, 2023.

Juncheng Wu, Wenlong Deng, Xingxuan Li, Sheng Liu, Taomian Mi, Yifan Peng, Ziyang Xu, Yi Liu, Hyunjin Cho, Chang-In Choi, et al. Medreason: Eliciting factual medical reasoning steps in llms via knowledge graphs. *arXiv preprint arXiv:2504.00993*, 2025.

Jin Ye, Guoan Wang, Yanjun Li, Zhongying Deng, Wei Li, Tianbin Li, Haodong Duan, Ziyan Huang, Yanzhou Su, Benyou Wang, et al. Gmai-mmbench: A comprehensive multimodal evaluation benchmark towards general medical ai. *Advances in Neural Information Processing Systems*, 37: 94327–94427, 2024.

Xiang Yue, Yuansheng Ni, Kai Zhang, Tianyu Zheng, Ruoqi Liu, Ge Zhang, Samuel Stevens, Dongfu Jiang, Weiming Ren, Yuxuan Sun, et al. Mmmu: A massive multi-discipline multimodal understanding and reasoning benchmark for expert agi. In *Proceedings of the IEEE/CVF Conference on Computer Vision and Pattern Recognition*, pp. 9556–9567, 2024.

Aohan Zeng, Xin Lv, Qinkai Zheng, Zhenyu Hou, Bin Chen, Chengxing Xie, Cunxiang Wang, Da Yin, Hao Zeng, Jiajie Zhang, et al. Glm-4.5: Agentic, reasoning, and coding (arc) foundation models. *arXiv preprint arXiv:2508.06471*, 2025.

Sheng Zhang, Yanbo Xu, Naoto Usuyama, Hanwen Xu, Jaspreet Bagga, Robert Tinn, Sam Preston, Rajesh Rao, Mu Wei, Naveen Valluri, et al. Biomedclip: a multimodal biomedical foundation model pretrained from fifteen million scientific image-text pairs. *arXiv preprint arXiv:2303.00915*, 2023a.

Xiaoman Zhang, Chaoyi Wu, Ziheng Zhao, Weixiong Lin, Ya Zhang, Yanfeng Wang, and Weidi Xie. Pmc-vqa: Visual instruction tuning for medical visual question answering. *arXiv preprint arXiv:2305.10415*, 2023b.

Juexiao Zhou, Xiaonan He, Liyuan Sun, Jiannan Xu, Xiuying Chen, Yuetan Chu, Longxi Zhou, Xingyu Liao, Bin Zhang, Shawn Afvari, et al. Pre-trained multimodal large language model enhances dermatological diagnosis using skingpt-4. *Nature Communications*, 15(1):5649, 2024.

Yuxin Zuo, Shang Qu, Yifei Li, Zhangren Chen, Xuekai Zhu, Ermo Hua, Kaiyan Zhang, Ning Ding, and Bowen Zhou. Medxpertqa: Benchmarking expert-level medical reasoning and understanding. *arXiv preprint arXiv:2501.18362*, 2025.

APPENDIX

## A   THE USE OF LARGE LANGUAGE MODELS (LLMs)

During the preparation of this manuscript, we used OpenAI's GPT-5 model for minor language refinement and smoothing of the writing. The AI tool was not used for generating original content, conducting data analysis, or formulating core scientific ideas. All conceptual development, experimentation, and interpretation were conducted independently without reliance on AI tools.

## B   REPRODUCIBILITY STATEMENT

We provide detailed instruction to reproduce our research. The prompts for generation and verification are presented in Section E. Section 3 provides instructions for initial data curation and filtering, question generation and verification, and our implementation details. Section 4.1 demonstrates the experimental settings, including benchmarks and comparison baselines. All models are trained on 8 A100 GPU machines using mixed precision. The hyperparameters are following "Med-VLThinker" (Huang et al., 2025b). We full-stack release our data, code, and models in the pursuit of open science [1].

## C   ETHICAL CONSIDERATIONS AND LICENSING

MedVLSynther operates solely on PMC-OA literature. We include source metadata and original licenses in dataset manifests and exclude items whose licenses are incompatible with redistribution for research. We provide attribution to source articles in per-item metadata. MedVLSynther is intended exclusively for research; no clinical use is authorized. We release prompts, rubric, and filtering logs to enable community auditing.

## D   MORE DATA STATISTICS

Table 7: Image secondary label from original Biomedica statistics.

| Image Secondary Label | # Images | Ratio | Image Secondary Label | # Images | Ratio |
|---|---|---|---|---|---|
| clinical imaging | 1039 | 7.17 | intraoral imaging | 393 | 2.71 |
| brain | 984 | 6.79 | eye | 379 | 2.62 |
| x-ray radiography | 879 | 6.07 | electrocardiography | 377 | 2.60 |
| light microscopy | 829 | 5.72 | mammography | 374 | 2.58 |
| computerized tomography | 785 | 5.42 | patient photo | 357 | 2.46 |
| immunohistochemistry | 676 | 4.67 | procedural image | 335 | 2.31 |
| magnetic resonance | 673 | 4.65 | laryngoscopy | 314 | 2.17 |
| surgical procedure | 565 | 3.90 | microscopy | 65 | 0.45 |
| skin lesion | 563 | 3.89 | scientific illustration | 31 | 0.21 |
| functional magnetic resonance | 525 | 3.62 | line plot | 27 | 0.19 |
| angiography | 519 | 3.58 | bar plot | 23 | 0.16 |
| intraoperative image | 504 | 3.48 | immunoblot | 12 | 0.08 |
| specimen | 500 | 3.45 | confocal microscopy | 11 | 0.08 |
| optical coherence tomography | 492 | 3.40 | aerial photography | 2 | 0.01 |
| ultrasound | 486 | 3.35 | table | 2 | 0.01 |
| endoscopy | 477 | 3.29 | tree | 1 | 0.01 |
| teeth | 470 | 3.24 | signal plot | 1 | 0.01 |
| skull | 415 | 2.86 | user interface | 1 | 0.01 |
| plot | 400 | 2.76 | ambiguous | 1 | 0.01 |

---

[1]https://ucsc-vlaa.github.io/MedVLSynther/

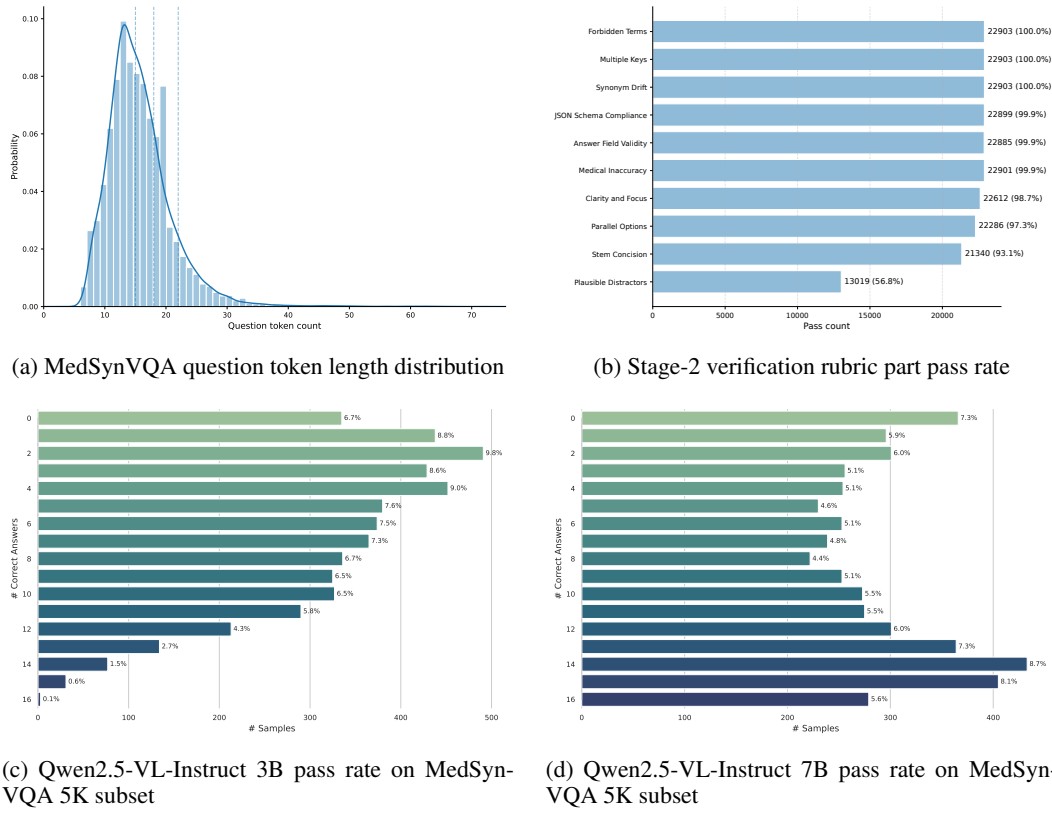

(a) MedSynVQA question token length distribution

(b) Stage-2 verification rubric part pass rate

(c) Qwen2.5-VL-Instruct 3B pass rate on MedSyn-VQA 5K subset

(d) Qwen2.5-VL-Instruct 7B pass rate on MedSyn-VQA 5K subset

Figure 5: More data statistics about MedSynVQA

## E  PROMPTS

---

**Generator Prompt**

[SYSTEM ROLE]
You are an expert medical-education item writer. Your job is to generate a high-quality multiple-choice question (MCQ) from a biomedical figure (the image) and its accompanying paper caption/context. The MCQ must be self-contained, clinically valid, and solvable by carefully inspecting the image together with general domain knowledge implicitly derivable from the caption/context: WITHOUT quoting the caption or revealing the answer verbatim.
[NON-NEGOTIABLE RULES]
1) Do NOT write "according to the caption/description/text" or similar.
2) Do NOT copy answer text verbatim from the caption; paraphrase and compress.
3) Exactly ONE best answer. Distractors must be plausible and mutually exclusive with the key.
4) Use only information supported by the image and facts that a competent clinician could infer from the caption/context; no speculative claims.
5) Keep clinical terminology precise; avoid brand names/PHI; no patient identifiers.
6) Output MUST follow the JSON schema below-no extra keys, no commentary.
7) Do NOT include chain-of-thought. Keep rationales concise and factual is NOT required-only output the JSON.
[RUBRIC (Self-check before you output)]
Essential Criteria (must PASS)
- E1. Stem is self-contained; no mention of "caption/description"; no hidden assumptions.
- E2. Image-content alignment: the question requires inspecting specific visual features.

---

- E3. Caption-derived facts are integrated implicitly (paraphrased) but do not leak the answer.
- E4. Single correct option; remaining options are incorrect for a clear clinical reason.
- E5. Medical correctness: terminology, anatomy, modality, and pathophysiology are accurate.
Important Criteria (strongly recommended)
- I1. Cognitive level: $\geq$ application (identification, interpretation, next step, best explanation).
- I2. Distractors: near-misses, common confusions, or plausible alternatives-not trivial.
- I3. Parallelism: options have similar length/structure; avoid "all/none of the above."
- I4. Difficulty labeled (Easy/Moderate/Hard) with a brief justification. (Do NOT include this in the output JSON.)
- I5. Balanced scope: focuses on one primary concept (finding, diagnosis, step, location).
Optional Criteria (nice to have)
- O1. Localizes the key finding (e.g., lobe/segment/organ subregion) if appropriate.
- O2. Uses quantitative details (size/scale/grade/stage) only when clearly supported.
[ALLOWED QUESTION ARCHETYPES] Pick ONE that best fits the image + derivable facts:
- Finding identification ("Which abnormality is present?")
- Best diagnosis / most likely explanation
- Next best step (diagnostic or management)
- Localization ("Which structure/region is affected?")
- Modality/sequence recognition (e.g., T1 vs T2, phase, stain)
[GENERATION WORKFLOW]
Step 1 - (Privately) derive a few concise, paraphrased facts from captions/contexts that a clinician could reasonably infer.
Step 2 - Choose an archetype so that both image inspection and those facts are needed.
Step 3 - Write a self-contained stem that integrates the derived facts implicitly (no mention of "caption/description/text") and does NOT reveal the answer.
Step 4 - Author 5 options (A-E): one correct, four high-quality distractors (near-miss/opposite/irrelevant-but-plausible). Keep options parallel.
Step 5 - Run the RUBRIC. If any Essential item fails, regenerate (internally). Output only JSON.
[OUTPUT FORMAT - STRICT JSON]
Return exactly one JSON object with keys:
- "question": string
- "options": object with keys "A","B","C","D","E"
- "answer": one of "A","B","C","D","E"
[SOURCE MATERIAL]
CAPTIONS: {CAPTION_BLOCK}
CONTEXTS: {CONTEXT_BLOCK}
Think silently. Output ONLY the JSON object.

---

### Verifier Prompt

[SYSTEM ROLE]
You are a biomedical MCQ verifier with two distinct roles, executed in order:
1. **The Referee (for Essential Criteria):** You are an objective, rule-based official. Your job is to fairly determine if the absolute minimum standards for usability are met.
2. **The Critic (for Bonus Criteria):** After the essential check, you become a relentless perfectionist. Your default assumption is that the MCQ is NOT excellent. Your goal is to deny bonus points unless perfection is demonstrated beyond any doubt.
Judge each MCQ *only* using FIGURE(s)+CAPTION+CONTEXT.
[DATASET POLICY - MUST ENFORCE]
- Stem is self-contained and MUST NOT say "caption" or "context".

- Paraphrasing allowed, but DO NOT introduce unsupported clinical facts (age/sex/history/location/findings/diagnoses) beyond CAPTION/CONTEXT or clearly visible image cues.
- If CAPTION names a diagnosis, do NOT restate that diagnosis verbatim in the stem.
- Exactly one correct option; all options must be the same semantic type.
- Clinical correctness (modality/stain/anatomy/pathophysiology) must be supported by sources.
[SCORING MODEL] Output ONLY "rubric":[...].
Each item has:
- idx (1..N), title, description (ONE sentence starting with "Essential/Important/Optional/Pitfall Criteria: ..."),
- category: Essential |Important |Optional |Pitfall,
- weight: Essential=5; Important in 3,4; Optional in 1,2; Pitfall in -1,-2,
- score: Essential/Important/Optional in 0, weight; Pitfall in 0, weight (0 = not triggered),
- notes: $\leq$ 12 words (brief reason; no chain-of-thought).
**Scoring Mindset:**
- **Essential Items:** Award full points if the rule is met. Be a fair and impartial referee.
- **Important/Optional Items (Bonus Points):** **These are bonus points, not entitlements.** The score is **0 by default.** You must find **irrefutable evidence of perfection** to award points. If there is *any* subjective room for improvement, the score remains 0.
[FIXED ESSENTIAL ITEMS - MUST INCLUDE with EXACT titles]
1) Stem Self-contained
2) Vocabulary Constraint
3) Diagnosis Leak
4) Single Correct Option
5) Option Type Consistency
6) Clinical Validity
7) Image–Text Consistency
[BONUS CRITERIA FOR EXCELLENCE - ZERO-TOLERANCE JUDGEMENT]
(You must assess against a diverse set of 4-8 items. For this section, your mindset is "guilty until proven innocent." **A single, minor flaw in any sub-point means an instant score of 0 for the entire item.**)
- Plausible Distractors (Important, 3-4)
- **Every single** distractor must be a strong, clinically relevant alternative given the sources.
- Each must differ from the key by exactly ONE clear axis.
- **If you can imagine a slightly more plausible distractor that wasn't used, score 0.** There must be no weak links.
- Parallel Options (Important, 3)
- Grammatical structure, length, and specificity must be **rigorously uniform**.
- **Any noticeable outlier** in form (e.g., one starts with a verb, others with nouns) or length fails this. No unique cues on the key.
- Stem Concision (Optional, 1-2)
- Stem must be $\leq$ 2 sentences AND $\leq$ 35 words.
- If the stem can be rephrased to be even **slightly more elegant or direct** without losing critical meaning, **score 0**.
- Clarity and Focus (Optional, 2)
- The stem poses a single, perfectly unambiguous question.
- If the question could be worded **any more clearly or is even slightly awkward**, **score 0**.
- Answer Field Validity (Important, 3)
- 'answer' exists, is one of A-E, and exactly matches an option key.
- No duplicate options; all option strings non-empty.
- JSON Schema Compliance (Important, 3)
- MCQ has exactly required keys question, optionsA..E, answer; no extras/missing.
- Forbidden Terms (Pitfall, -2)
- Stem contains the exact word 'caption' or 'context'.

- Synonym Drift (Pitfall, -1)
- Stem introduces a **specific** clinical fact that is absent from sources and not visible in the image.
- Multiple Keys (Pitfall, -2)
- >1 option is reasonably correct **given sources**. - Medical Inaccuracy (Pitfall, -2)
- Any statement directly contradicts sources.
[HOW TO JUDGE]
- For Essentials, be a referee. For the Bonus section, be a rival looking for a weakness.
- Use only FIGURE(s)+CAPTION+CONTEXT. If an MCQ asserts anything unsupported, fail the relevant item.
[OUTPUT FORMAT — STRICT JSON ONLY]
Return exactly: {"rubric": [{"idx":1,"title":"...","description":"Essential Criteria: ...","category":"Essential","weight":5,"score":0 or 5,"notes":"..."},...]}
No totals. No extra keys. No commentary.
[INPUTS]
IMAGES: <attach in order >
MCQ: {MCQ_JSON}
CAPTION: {CAPTION_BLOCK}
CONTEXT: {CONTEXT_BLOCK}
[CONDUCT]
If inputs are insufficient to judge, output {"error":"insufficient_evidence"} ONLY. Think silently. Output ONLY the JSON object above.

## F  DATA CONTAMINATION ANALYSIS

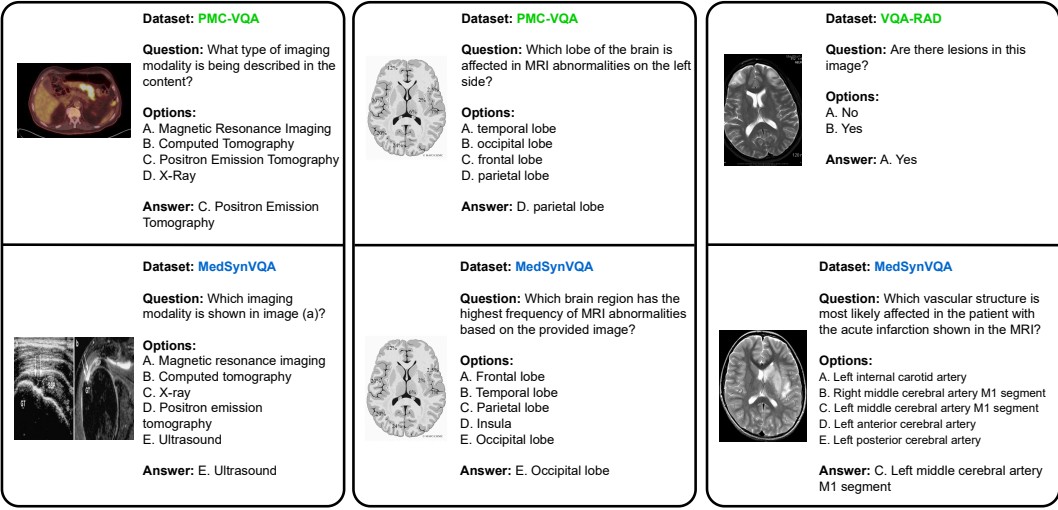

Figure 6: Examples from our contamination analysis across MedSynVQA and our testset. On the left, this case shows that top-10 text-embedding neighbor shows similar options but entirely different images. In the middle, we find no exact duplicates by MD5; pHash retrieves a same-source size variant with different question/answers. On the right, image-embedding neighbor reflects modality clustering (high similarity) without instance identity; questions also differ.

**Text decontamination.** We audit overlap between the training pool (13,087 rows) and the held-out test set (8,220 rows) over the fields question and options, after lowercasing, whitespace cleanup, digit masking (all numbers to "<NUM>"), and normalizing options to "A./B./...". 1) MinHash + Levenshtein hashes 3-character n-grams and confirms candidates with normalized edit similarity ($\tau = 0.90$, following Pal et al. (2022)). This conservative pass finds no near-duplicate strings

(pairs=0; hit queries=0; hit rate=0.0). 2) Embedding + FAISS (Douze et al., 2024) encodes text with BAAI/bge-m3 (Chen et al., 2024a) (L2-normalized), retrieves top-k=5 train neighbors per test item, and flags pairs with cosine larger then 0.88 (Tumre et al., 2025). This yields 47 pairs across 23 test queries (hit_rate about 0.280%), with per-query hits mean=2.0 (max=5). The top-1 similarity for those 23 queries averages 0.899 (median 0.894; p95=0.926), and qualifying pair similarities concentrate in [0.88, 0.95). Considering all test items, the best-neighbor ("MaxSim") distribution is mean 0.651 (median 0.644; p95=0.778), and Overlap@0.88 = 23/8,220 = 0.280%, indicating minimal distributional leakage beyond trivial lexical matches. We found **no** overlap of texts.

**Image decontamination.** We repeat the two-pass audit on images decoded directly from the dataset (using the first image per record by default). 1) A hashing pipeline detects exact and near duplicates via MD5 of raw pixels and a 64-bit perceptual hash (pHash); pHash neighbors are retrieved with a binary FAISS index, and we summarize each test image's minimum Hamming distance and "Overlap@d" (fraction with best neighbor less than or equal to "d"). 2) A semantic pipeline encodes images with OpenCLIP or BiomedCLIP (Ilharco et al., 2021; Zhang et al., 2023a), searches the train set with FAISS (Douze et al., 2024) (inner product on L2-normalized features), and reports per-test MaxSim, Overlap@$\tau$, and a histogram of high-similarity test–train pairs (e.g., at $\tau = 0.88$). Together, hashing probes pixel-level or layout-level duplication while embeddings probe semantic or style reuse; all statistics are computed from the images as stored in parquet under this single-image-per-record protocol. We found **no** overlap of images.

## G    EXPERT EVALUATION STATISTICS

To complement the downstream results and decontamination analysis, we conducted a small-scale human evaluation of MedSynVQA questions to assess intrinsic item quality and the behavior of the verifier. **Five** board-certified imaging experts participated, with clinical backgrounds covering oncologic CT/MR, abdominal and gastrointestinal CT/MR, brain CT/MR, diagnostic radiology, and gastrointestinal endoscopy. Some questions were jointly reviewed by more than one expert when they spanned multiple organs or modalities.

### G.1    VERIFIER-ACCEPTED ITEMS

We first assessed the precision of the verifier on items it accepts. We randomly sampled 15 multiple-choice questions (MCQs) that had been accepted under the main verification threshold. Each item was rated on four 1 to 4 Likert scales (4 = excellent, 1 = poor): medical correctness and uniqueness of the keyed answer, clarity and professional wording, image grounding, and option design. In addition, experts provided a binary judgment on whether the item was acceptable for use as a medical MCQ.

Table 8 summarizes the aggregate statistics, and  10 reports per-item ratings. 14 of the 15 items (**about 93%**) were judged acceptable overall; one item was rejected because the brain MR image had such poor contrast that the expert considered the image non-evaluable, independent of the question text. The mean scores across criteria were 3.35 for medical correctness, 3.13 for clarity and professional wording, 3.46 for image grounding, and 3.40 for option design, corresponding to good to excellent quality on average. Two items received a score of 1 on the clarity and wording scale, indicating that while their medical content and image use were sound, the phrasing could be substantially improved. This is consistent with our finding that the current vocabulary related checks in the verifier do not fully capture wording quality and suggests that prompt refinement for clarity is a promising direction.

### G.2    VERIFIER-REJECTED ITEMS

We then examined whether the verifier is justified when rejecting items. We randomly sampled 20 MCQs that had been filtered out by the verifier and, for each item, selected one essential rubric that had received a zero score. The audited rubrics and their frequencies, along with human agreement, are summarized in Table 9; Table 11 provides per-item details.

Across the 20 rejected items, the audited essentials were: stem self-contained (5 items), vocabulary constraint (4), diagnosis leak (1), single correct option (2), option type consistency (4), clinical validity (2), and image–text consistency (2). For rubric types that do not require strong medical expertise

(stem self-contained, vocabulary constraint, option type consistency), the authors first inspected the items. Stems marked as non self-contained were indeed unusable without external context, for example explicitly referring to "according to the context" rather than being answerable from the stem and image alone. Option type consistency failures involved heterogeneous option formats or options literally written as "none" (for example, option E is just "none" as an empty placeholder, rather than a meaningful choice such as "no abnormality" or "none of the above"), so these rejections were appropriate.

For the vocabulary constraint rubric, we identified a systematic issue. In the detailed cases examined, the verifier incorrectly flagged sex, age, or clinical history as "unsupported," even though these attributes were present in the caption or context. These are genuine false negatives: the current rubric prompt does not reliably capture linguistic professionalism or terminology use and can misfire when the model fails to correctly read the context. We therefore regard vocabulary constraint as a weaker signal and a primary target for future prompt and rubric refinement.

For the medically heavy essentials (diagnosis leak, single correct option, clinical validity, and image-text consistency), items were evaluated by clinicians. In all inspected cases, experts agreed with the verifier. Items rejected for diagnosis leak did reveal the diagnosis directly in the stem; items rejected for single correct option or clinical validity had multiple plausible keys or medically incorrect statements; items rejected for image-text consistency did not match the visual evidence.

## G.3 SUMMARY

Although this human study is necessarily small due to limited clinician time and the difficulty of reviewing full image-based MCQs, it already involves five specialists and includes items that required cross-domain input. The audit shows that a large majority of verifier-accepted items are judged medically sound and usable, and that the essential medical rubrics driving rejection (diagnosis leak, single correct option, clinical validity, image–text consistency) align well with clinician judgment. The main systematic discrepancy is concentrated in the vocabulary-related check, which we now explicitly highlight as a limitation. Overall, Tables 8–11 provide intrinsic evidence about the reliability and failure modes of the verification stage and complement the downstream metrics and decontamination analyses in the main paper.

Table 8: Summary of expert evaluation on questions.

| Subset | Items | Experts | Human pass rate | Mean medical correctness | Mean clarity | Mean image grounding | Mean option design |
|---|---|---|---|---|---|---|---|
| Verifier-accepted MCQs | 15.00 | 5.00 | 14 / 15 (93%) | 3.35 | 3.13 | 3.46 | 3.40 |
| Verifier-rejected MCQs | 20.00 | 5.00 | N/A | N/A | N/A | N/A | N/A |

Table 9: Distribution of audited essential rubrics for rejected items.

| Essential rubric | Count | Human agreement with verifier | Notes |
|---|---|---|---|
| Stem self-contained | 5 | Yes (all) | Stems depended on external context, not answerable from stem+image |
| Vocabulary constraint | 4 | No (in 3 cases) | Verifier wrongly flagged sex/age/history as unsupported |
| Diagnosis leak | 1 | Yes (all) | Diagnosis explicitly revealed in stem |
| Single correct option | 2 | Yes (all) | Multiple plausible keys or ambiguous answer |
| Option type consistency | 4 | Yes (all) | Heterogeneous option formats or meaningless "none" placeholder |
| Clinical validity | 2 | Yes (all) | Medically incorrect or inconsistent statements |
| Image–text consistency | 2 | Yes (all) | Text description did not match the visual findings |

Table 10: Per-item expert ratings for verifier-accepted MCQs.

| Item ID | Reviewer specialty | Human acceptable? | Medical correctness (1–4) | Clarity & wording (1–4) | Image grounding (1–4) | Option design (1–4) | Notes (optional) |
|---|---|---|---|---|---|---|---|
| A01 | diagnostic radiology | Yes | 3 | 3 | 4 | 3 | |
| A02 | diagnostic radiology | Yes | 4 | 3 | 4 | 3 | |
| A03 | diagnostic radiology | Yes | 4 | 4 | 4 | 4 | |
| A04 | diagnostic radiology | Yes | 4 | 4 | 4 | 4 | |
| A05 | diagnostic radiology | Yes | 4 | 4 | 4 | 4 | |
| A06 | diagnostic radiology | Yes | 3 | 3 | 3 | 3 | |
| A07 | gastrointestinal endoscopy | Yes | 4 | 4 | 4 | 4 | |
| A08 | gastrointestinal endoscopy | Yes | 3 | 3 | 3 | 3 | |
| A09 | gastrointestinal endoscopy | Yes | 4 | 3 | 3 | 3 | |
| A10 | gastrointestinal endoscopy | Yes | 2 | 3 | 2 | 3 | |
| A11 | brain CT/MR | Yes | 4 | 4 | 4 | 4 | |
| A12 | brain CT/MR | No | N/A | 1 | 3 | 3 | poor contrast images |
| A13 | brain CT/MR | Yes | 4 | 4 | 4 | 4 | |
| A14 | brain CT/MR | Yes | 2 | 3 | 2 | 4 | |
| A15 | brain CT/MR | Yes | 2 | 1 | 4 | 2 | |

Table 11: Per-item audit for verifier-rejected MCQs.

| Item ID | Reviewer specialty | Essential rubric checked | Human agrees with rejection? |
|---|---|---|---|
| B01 | author self-review | Stem self-contained | Yes |
| B02 | author self-review | Stem self-contained | Yes |
| B03 | author self-review | Stem self-contained | Yes |
| B04 | author self-review | Stem self-contained | Yes |
| B05 | author self-review | Stem self-contained | Yes |
| B06 | author self-review | vocabulary constraint | Yes |
| B07 | author self-review | vocabulary constraint | No |
| B08 | author self-review | vocabulary constraint | No |
| B09 | author self-review | vocabulary constraint | No |
| B10 | oncologic CT/MR, diagnostic radiology | diagnosis leak | Yes |
| B11 | oncologic CT/MR | single correct option | Yes |
| B12 | gastrointestinal CT/MR, gastrointestinal endoscopy | single correct option | Yes |
| B13 | author self-review | option type consistency | Yes |
| B14 | author self-review | option type consistency | Yes |
| B15 | author self-review | option type consistency | Yes |
| B16 | author self-review | option type consistency | Yes |
| B17 | gastrointestinal CT/MR, gastrointestinal endoscopy | clinical validity | Yes |
| B18 | gastrointestinal CT/MR, gastrointestinal endoscopy | clinical validity | Yes |
| B19 | gastrointestinal CT/MR, gastrointestinal endoscopy | image-text consistency | Yes |
| B20 | brain CT/MR | image-text consistency | Yes |

# H CASE STUDY OF SYNTHETIC DATA

---

**Essential Criteria: Stem Self-contained**

**Generated Question:** Which finding is most consistent with the severe osteolysis described in the context?

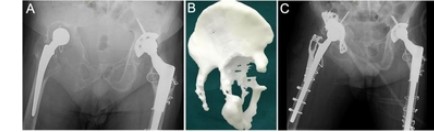

**Generated Choices:** (A) Intact ischial ramus on AM model (B) Complete destruction of ischial ramus (C) Stable acetabular prosthesis (D) Minimal bone loss around acetabulum (E) Intact pubic ramus on AM model

**Generated Correct Answer:** (B)

**Verifier Key Output:**
{ "idx": 1, "title": "Stem Self-contained", "description": "Essential Criteria: Stem does not rely on caption or context.", "category": "Essential", "weight": 5, "score": 0, "notes": "Stem references 'context', violating self-containment." },

---

**Analysis:**
*This item should fail with score 0 on Stem Self-contained. The phrase in the context makes the answer contingent on external text; without that context, the solver cannot infer that severe osteolysis corresponds to complete destruction of the ischial ramus, so option B is not reachable from the stem alone. The abbreviation AM model is also undefined in the stem, further harming self-containment.*

---

---

**Essential Criteria: Vocabulary Constraint**

**Caption**: A six-year-old girl with right side neglected modified Severin classification type IV developmental dysplasia of hip.

**Context**: A 6-year-old girl was admitted to our outpatient clinic with complaints of limping in June 2009...

**Generated Question**: Which abnormality is most consistent with the radiographic findings in this six-year-old girl?

**Generated Choices:** (A) Right hip subluxation with acetabular dysplasia (B) Left hip dislocation with femoral head avascular necrosis (C) Bilateral femoral neck fractures (D) Pelvic osteomyelitis (E) Sacroiliac joint ankylosis

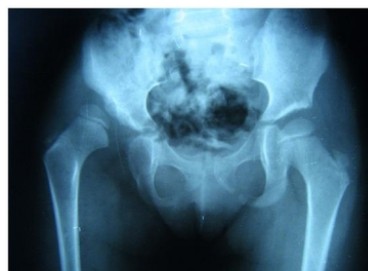

**Generated Correct Answer:** (A)

**Verifier Key Output:**
{ "idx": 2, "title": "Vocabulary Constraint", "description": "Essential Criteria: No unsupported clinical facts introduced.", "category": "Essential", "weight": 5, "score": 0, "notes": "Stem introduces age and sex not visible in image." },

---

**Analysis:**
*This item should receive score 0 on Vocabulary Constraint. The stem introduces age and sex as visible facts, while the evaluation setting treats the image as the sole evidence; these demographics are not directly verifiable from the image and are not required to answer. Including them constitutes unsupported clinical facts and also injects bias by priming pediatric diagnoses such as developmental dysplasia of the hip, reducing the emphasis on visual reasoning. This is one relatively acceptable example for vocabulary constraint rejection.*

---

**Essential Criteria: Diagnosis Leak**

**Caption**: A 37-year-old woman with a palpable mass on the upper outer quadrant of the right breast. (a) Cranio-caudal mammogram. (b) Mediolateral Oblique mammogram. Arrows in (a and b) mark the lesion. The triangular skin marker indicating the location of the palpable mass is visible in (b). (c-e) Ultrasound imaging show a circumscribed $1.6 \times 1.8 \times 1.5$ cm hypoechoic mass with flow and post enhancement at 11 o'clock position and 10 cm from nipple corresponding to the mass in mammograms. (f) Ultrasound image from biopsy procedure showing the needle. (g) invasive

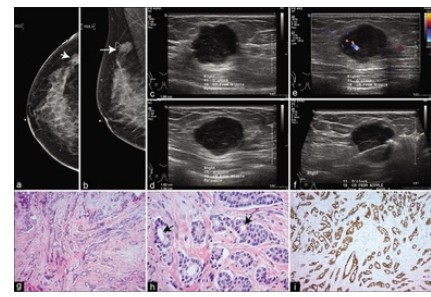

ductal carcinoma, grade 1, showing infiltrating malignant glands within desmoplastic stroma (H&E stain). (h) high-power magnification showing predominant tubular formation (arrows), mild nuclear atypia with rare/no mitosis (H&E stain). (i) ER immunostain showing strong diffuse positivity with anti-ER antibody.

**Context**: Tumor differentiation graded on an ordinal scale (3: poorly-differentiated, 2: moderately-differentiated, and 1: well-differentiated) was positively correlated with US measured tumor size...

**Generated Question**: A 37-year-old woman presents with a palpable mass in the upper outer quadrant of the right breast. Imaging and biopsy reveal a circumscribed hypoechoic

---

mass with flow, invasive ductal carcinoma, grade 1, with predominant tubular formation, mild nuclear atypia, and strong diffuse ER positivity. Which histopathological feature is most consistent with the tumor grade?

**Generated Choices:** (A) Poorly differentiated with marked nuclear atypia and frequent mitoses (B) Moderately differentiated with solid nests and moderate nuclear atypia (C) Well-differentiated with predominant tubular formation and mild nuclear atypia (D) In situ carcinoma with comedonecrosis (E) Metastatic carcinoma with signet ring cells

**Generated Correct Answer:** (C)

**Verifier Key Output:**
{ "idx": 3, "title": "Diagnosis Leak", "description": "Essential Criteria: The stem does not restate the diagnosis verbatim.", "category": "Essential", "weight": 5, "score": 0, "notes": "Stem restates diagnosis from caption." },

---

**Analysis:**
*This item exhibits clear diagnosis leakage. The stem reproduces the caption's diagnostic content verbatim, namely invasive ductal carcinoma, grade 1, predominant tubular formation, mild nuclear atypia, and strong ER positivity. Option C mirrors these phrases, so a question only baseline can answer without using the image, producing a direct stem to option match. This dual overlap, caption to stem and stem to option, invalidates the assessment.*

## Essential Criteria: Single Correct Option

**Caption**: Panoramic view after implant insertion.
**Context**: None
**Generated Question**: Which of the following best describes the radiographic appearance of the dental implants in this panoramic view?

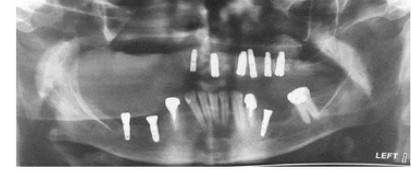

**Generated Choices:** (A) All implants are fully osseointegrated with no peri-implant radiolucency (B) Multiple implants show peri-implant radiolucency indicating early failure (C) Implants are positioned in the maxillary arch (D) Implants are surrounded by dense cortical bone (E) Implants exhibit significant angulation errors

**Generated Correct Answer:** (B)

**Verifier Key Output:**
{ "idx": 4, "title": "Single Correct Option", "description": "Essential Criteria: Only one option is correct based on sources.", "category": "Essential", "weight": 5, "score": 0, "notes": "Option B is not clearly supported by image." },

---

**Analysis:**
*With a panoramic radiograph of limited resolution, option B asserts both the presence of peri-implant radiolucency around multiple implants and the clinical inference of early failure. The first claim is not clearly visible on this single image, and the second cannot be concluded from imaging alone. Option A, "no peri-implant radiolucency," is likewise non-falsifiable here and may seem plausible if details are indistinct, creating an A vs B ambiguity. Options D and E are subjective without quantitative thresholds, and while C is clearly incorrect, its presence does not make any remaining option uniquely correct. The net effect is that more than one answer can be defended from the same image.*

---

**Essential Criteria: Option Type Consistency**

**Caption**: Scoliosis series and kyphosis series images of the patient. The scoliosis anteroposterior view image (2A) shows the patient's trunk flexed to the left side by 12.31 degrees; the kyphosis series lateral image (2B) shows anterior flexion by 16.18 degrees of the patient's trunk.

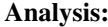

**Context**: None

**Generated Question**: Which measurement represents the degree of trunk flexion in the lateral view?

**Generated Choices:** (A) 12.31 degrees (B) 16.18 degrees (C) Anterior flexion (D) Left-side flexion (E) Trunk flexion angle

**Generated Correct Answer:** (B)

**Verifier Key Output:**
{ "idx": 5, "title": "Option Type Consistency", "description": "Essential Criteria: All options are the same semantic type.", "category": "Essential", "weight": 5, "score": 0, "notes": "Options mix numeric values and descriptive terms." },

---

**Analysis:**
*The stem asks for "which measurement represents the degree of trunk flexion in the lateral view," so the expected answer type is a numeric value. The options, however, mix numbers and descriptive phrases: A and B are numeric, whereas C, D, and E are textual descriptors. This type mismatch violates the requirement that all options be of the same semantic class and also weakens alignment with the stem's focus on a measurement.*

---

**Essential Criteria: Clinical Validity**

**Caption**: A 41-year-old male with cervical tuberculosis suffered from neck pain, spastic quadriparesis, and dysphagia due to a retropharyngeal abscess. Neutral lateral cervical spine radiograph (A) indicates cervical kyphosis of 51 degree at C3-6, and C4 is the apical vertebra. The kyphosis Cobb angle was 54 dgree and 49 degree on flexion and extension, respectively (B, C), which indicated that the flexibility was only 3.9%. MRI showed the formation of a retropharyngeal abscess and ventral compression of the spinal cord. The Cobb angle was 21 degree on MRI, which indicated that the flexibility was 58.8% (D). The x-ray and CT scans at the final follow-up showed that the correction was well maintained without sig-

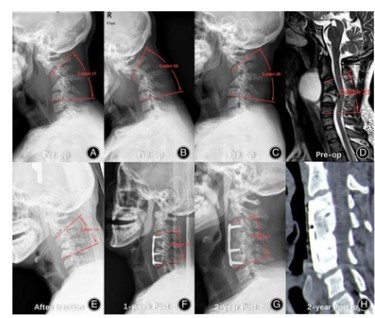

nificant loss, and solid bone fusion was finally achieved. **Context**: ...Radiological examinations (cervical X-ray, MRI, and CT) were performed to assess the bone fusion state, flexibility and correction of kyphosis. ... The flexibility of kyphosis was evaluated on both dynamic extension-flexion X-rays and cervical MRI ...

**Generated Question**: Which imaging modality best demonstrates the flexibility of cervical kyphosis in this patient?

**Generated Choices:** (A) Neutral lateral cervical spine radiograph (B) Flexion lateral cervical spine radiograph (C) Extension lateral cervical spine radiograph (D) MRI (E) CT scan

**Generated Correct Answer:** (D)

**Verifier Key Output:**

{ "idx": 6, "title": "Clinical Validity", "description": "Essential Criteria: Clinical correctness supported by sources.", "category": "Essential", "weight": 5, "score": 0, "notes": "MRI does not best demonstrate flexibility." },

---

**Analysis:**
*The question asks which modality best demonstrates the flexibility of cervical kyphosis, but standard assessment of flexibility is by dynamic flexion and extension lateral radiographs, which directly show angle change. MRI is primarily for soft-tissue and cord compression assessment, is usually obtained supine, and is not a dynamic study, so it is not the best modality to demonstrate flexibility. Although the case text reports a percentage derived from the difference between the neutral X-ray angle and an MRI angle, that does not make MRI the best demonstration of flexibility.*

# I   CASE STUDY OF MODEL OUTPUTS

## MedVLSynther-3B 5K output for MMMU-medical

**Question**: In the diagram below, the P wave represents ( ), the QRS segment represents ( ) and the T wave represents ( ).
**Choices:** (A) atrial depolarization, ventricular depolarization, ventricular repolarization
(B) ventricular repolarization, atrial depolarization, ventricular depolarization
(C) ventricular depolarization, ventricular repolarization, atrial depolarization
**Correct Answer:** (A)
**Model Answer:** (A)
**Model Response:**
<think>The P wave is typically associated with atrial depolarization, which is the initial depolarization of the atria. The QRS segment represents ventricular depolarization, which is the depolarization of the ventricles. The T wave is associated with ventricular repolarization, which is the process

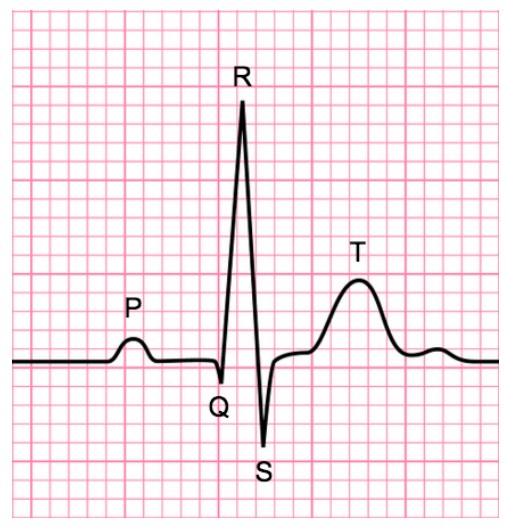

of the ventricles returning to their resting state after depolarization. Therefore, the correct sequence is P (atrial depolarization), QRS (ventricular depolarization), and T (ventricular repolarization). </think><answer>A. atrial depolarization, ventricular depolarization, ventricular repolarization</answer>

---

**Analysis:**
*The model correctly matches the P wave, QRS complex, and T wave with their respective physiological events: P wave for atrial depolarization, QRS complex for ventricular depolarization, and T wave for ventricular repolarization. However, its explanation could be enhanced by linking the morphology of these waves to their function. For instance, it could mention the small, rounded P wave representing atrial depolarization, the narrow, tall QRS complex signifying rapid ventricular depolarization (with atrial repolarization buried within), and the broader T wave for ventricular repolarization. Additionally, the model could have clarified the terminology, specifically that the "QRS complex" is the correct term, not the "QRS segment." Furthermore, while the explanation is correct, it would have been helpful to briefly exclude options B and C, which incorrectly describe the sequence of depolarization and repolarization.*

---

**MedVLSynther-3B 5K output for MedXpert-QA**

**Question**: When performing tension band fixation for an olecranon fracture, as illustrated in Figure A, which nerve is most vulnerable to injury if the Kirchner wire penetrates too deeply through the anterior cortex of the proximal ulna? **Choices:** (A) Median nerve
(B) Ulnar nerve
(C) Posterior interosseous nerve
(D) Radial nerve
(E) Anterior interosseous nerve
**Correct Answer:** (E)
**Model Answer:** (E)
**Model Response:**
<think>Olecranon fractures are located at the elbow joint, and the ulna is the bone that forms the elbow joint with the radius. The anterior cortex of the proximal ulna is where the Kirchner wire is typically placed

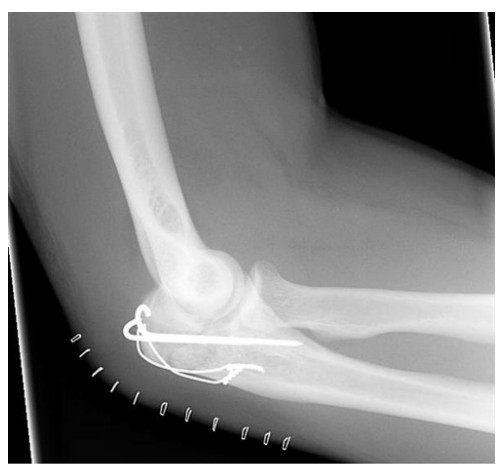

for tension band fixation. The anterior interosseous nerve runs along the anterior surface of the ulna, just below the anterior cortex. If the Kirchner wire penetrates too deeply through the anterior cortex, it could potentially injure the anterior interosseous nerve, which is the most vulnerable nerve in this location. </think><answer>E. Anterior interosseous nerve. </answer>

---

**Analysis:**
*The model demonstrates a solid grasp of surgical anatomy and spatial reasoning by correctly identifying the anterior interosseous nerve (AIN) as the nerve most at risk when the Kirschner wire penetrates too deeply through the anterior cortex of the proximal ulna during olecranon tension-band fixation. This shows that the model not only recalls factual anatomy but also understands the directional logic of the operative field—that danger arises from anterior cortical breach rather than posterior or medial misplacement. While its explanation slightly simplifies the anatomy, the overall reasoning is sound. The AIN indeed originates from the median nerve, passes deep to the pronator teres, and descends along the anterior interosseous membrane within the deep volar compartment—precisely the region a deeply driven wire could encroach upon. By contrast, the posterior interosseous nerve lies posterolaterally around the radial neck, the ulnar nerve remains protected posterior-medially within the cubital tunnel, and the median nerve trunk is more superficial and radial. These distinctions support the model's conclusion.*

## MedVLSynther-3B 5K output for PathVQA

**Question**: does microscopy show branching papillae having fibrovascular stalk covered by a single layer of cuboidal cells having ground-glass nuclei? **Choices:** (A) Yes (B) No

**Correct Answer:** (A)

**Model Answer:** (A)

**Model Response:**

<think>The question describes a histological feature that is characteristic of papillary thyroid carcinoma. Papillary thyroid carcinoma is a type of thyroid cancer that typically presents with branching papillae, which are structures that resemble branching tree-like structures. These papillae are covered by a single layer of cuboidal cells that have ground-glass nuclei. The presence of these features is consistent with the histological appearance of papillary thyroid carcinoma, which is often diagnosed through microscopic examination. </think><answer>A. Yes. </answer>

---

**Analysis:**

*The model accurately identifies the defining microscopic features of papillary thyroid carcinoma (PTC)—branching papillae with fibrovascular cores, lined by a single layer of cuboidal cells showing ground-glass ("Orphan Annie eye") nuclei—and correctly concludes "Yes." This demonstrates solid pattern recognition and understanding of the diagnostic morphology. While its reasoning leans toward a textbook summary rather than slide-specific description, the conceptual mapping is on point. It could be improved by explicitly referencing visible features such as the fibrovascular stalks, nuclear grooves, and intranuclear pseudoinclusions, as well as by differentiating from benign papillary hyperplasia or follicular variant PTC. Nonetheless, the model's response shows a commendable ability to link structural, cytologic, and diagnostic hallmarks coherently—an impressive level of interpretive accuracy for a 3B-scale pathology model.*

---

**MedVLSynther-3B 5K output for PMC-VQA**

**Question**: Which region of the brain is connected with the habenula that is highlighted in green? **Choices:** (A) Basal forebrain
(B) Periaqueductal gray
(C) Medial prefrontal cortex
(D) Raphe nuclei
**Correct Answer:** (C)
**Model Answer:** (C)
**Model Response:**

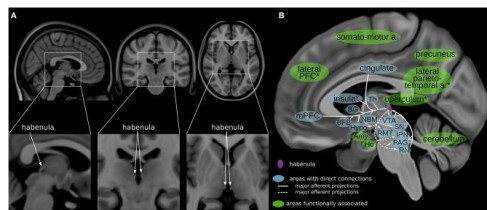

<think>The image shows a brain diagram with various regions labeled. The habenula is highlighted in green, and it is connected to several other regions. The green label next to the habenula points to the "medial prefrontal cortex," which is one of the areas associated with the habenula. This suggests that the medial prefrontal cortex is a region that is functionally associated with the habenula. </think><answer>C. Medial prefrontal cortex. </answer>

---

**Analysis:**
*The model correctly selects the medial prefrontal cortex (mPFC) as the region functionally associated with the habenula, demonstrating that it can interpret the visual cue of the green-highlighted region and map it to the correct brain area. This shows a sound grasp of color-coded neuroanatomical representations and context-based reasoning. However, its explanation remains somewhat superficial: it does not reference the figure legend, which clarifies that green indicates functional association while blue denotes direct anatomical connectivity, nor does it rule out the other options (basal forebrain, periaqueductal gray, raphe nuclei), which are blue rather than green in the diagram. Despite this, the reasoning pathway—linking color cue to functional correlation—is accurate and efficient. Overall, the model demonstrates solid spatial comprehension and context mapping, with room for improvement in precise figure interpretation and differential exclusion.*

**MedVLSynther-3B 5K output for SLAKE**

**Question**: Does the picture contain lung?
**Choices:** (A) Yes
(B) No
**Correct Answer:** (A)
**Model Answer:** (A)
**Model Response:**
<think>The image provided is a cross-sectional view of a human body, likely a CT scan. The visible structures include bones, soft tissues, and air-filled spaces. The air-filled spaces are characteristic of the lungs, which are responsible for gas exchange in the body. Therefore, the image does contain lung structures. </think><answer>A. Yes. </answer>

---

**Analysis:**
*The model correctly recognizes that the image contains lung parenchyma, identifying the low-attenuation, air-filled regions as pulmonary structures and noting the axial (cross-sectional) orientation of the scan. This indicates a proper understanding of how lung fields appear on CT—dark, symmetric areas surrounding the mediastinum. However, the explanation remains somewhat generic: it does not explicitly anchor its reasoning to clear anatomic landmarks such as the trachea, bronchi, ribs, or vertebral body, nor does it exclude other potential air-filled structures like bowel loops. A stronger rationale would mention the bilateral distribution of aerated areas within the thoracic cavity, bordered by ribs and mediastinum, confirming their identity as lungs rather than abdominal gas. Overall, the model shows correct interpretation and solid conceptual grounding, though finer spatial referencing and exclusion reasoning would further strengthen its diagnostic precision.*

---

**MedVLSynther-3B 5K output for VQA-RAD**

**Question**: Is there a small bowel obstruction? **Choices:** (A) Yes
(B) No
**Correct Answer:** (A)
**Model Answer:** (A)
**Model Response:**

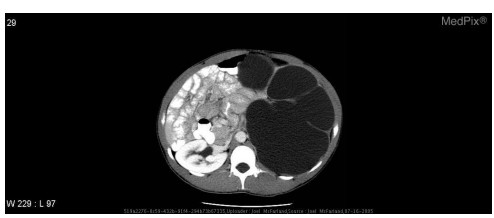

<think>A small bowel obstruction typically presents with signs of bowel dilation and fluid accumulation. In this CT scan, the small bowel loops appear dilated and there is a significant amount of fluid accumulation, which are characteristic signs of a small bowel obstruction. The presence of these findings strongly suggests the presence of a small bowel obstruction. </think><answer>A. Yes. </answer>

---

**Analysis:**
*The model correctly identifies small bowel obstruction (SBO) by recognizing the presence of bowel dilation and fluid accumulation, demonstrating a solid grasp of the key radiologic pattern. Its reasoning is directionally accurate and clinically valid, but somewhat generalized—it does not explicitly describe the hallmark CT features that confirm SBO. A stronger explanation would reference multiple centrally located dilated small-bowel loops (¿3 cm in diameter), numerous air–fluid levels, and possibly the small-bowel feces sign, contrasted with a collapsed distal colon. It would also mention the need to identify a transition point to confirm mechanical obstruction and distinguish it from paralytic ileus or large-bowel obstruction. Overall, the model's answer is correct and displays good diagnostic reasoning, though it could reach expert-level precision with more specific anatomic anchoring and differential exclusion.*

## MedVLSynther-7B 13K output for MMMU-medical

**Question**: This line of of myelinated axons in layer IV of visual cortex represents the axons of cells in the **Choices:** (A) Superior colliculus.
(B) Lateral geniculate.
(C) Retina.
(D) Medial geniculate.
**Correct Answer:** (B)
**Model Answer:** (B)
**Model Response:**

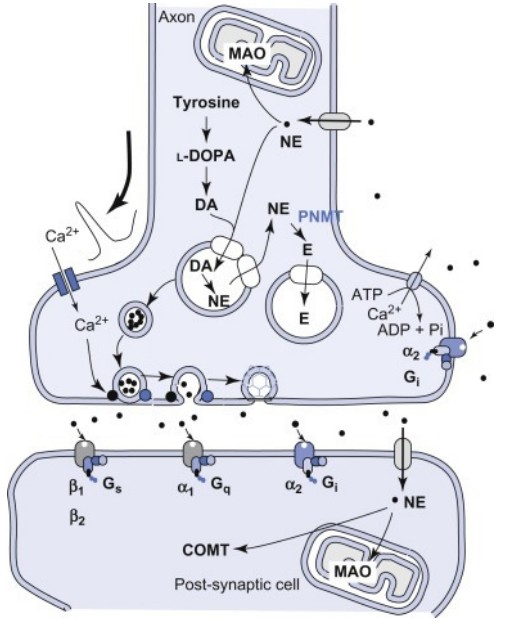

<think>The image shows a cross-section of the brain, specifically the visual cortex, which is part of the cerebral cortex. The layer IV in the visual cortex is known as the "cytoarchitectonic layer" and is characterized by the presence of large pyramidal cells. The myelinated axons in layer IV of the visual cortex are the axons of cells in the lateral geniculate nucleus (LGN), which is a relay nucleus in the visual pathway. The LGN receives input from the retina and sends it to the primary visual cortex (V1) for further processing.

Given the options: A. Superior colliculus - This is a part of the midbrain involved in visual and auditory processing, not the visual cortex. B. Lateral geniculate - This is the correct answer as the myelinated axons in layer IV of the visual cortex are the axons of cells in the lateral geniculate nucleus. C. Retina - The retina is the sensory organ that sends information to the LGN, not the LGN itself. D. Medial geniculate - This is a part of the auditory pathway, not the visual pathway.

Therefore, the correct answer is B. </think><answer>B. Lateral geniculate. </answer>

---

**Analysis:**
*The model correctly identifies that the myelinated axons in layer IV of the visual cortex originate from the lateral geniculate nucleus (LGN). Its reasoning demonstrates accurate knowledge of the visual pathway—that retinal ganglion cells project to the LGN, and LGN neurons, in turn, send their axons (optic radiations) to layer IV of the primary visual cortex (V1). The exclusion of other options (superior colliculus, retina, medial geniculate) is logically consistent and anatomically precise, showing solid understanding of sensory system segregation. The answer is well-grounded in neuroanatomy and pathway logic, though it could be further improved by explicitly linking the layer IV (stria of Gennari) feature to the LGN afferents and noting that the medial geniculate instead projects to auditory cortex. The model correctly traces the visual pathway, recognizing that myelinated axons in layer IV of the visual cortex originate from lateral geniculate nucleus neurons. Its reasoning is anatomically precise and well-structured, effectively excluding distractors from other sensory systems.*

## MedVLSynther-7B 13K output for MedXpertQA-MM

**Question**: What is the diagnosis? **Choices:**
(A) Echinococcal infection
(B) Pneumatosis intestinalis
(C) Trichobezoar
(D) Meckel''s diverticulitis
(E) Adrenal cancer
**Correct Answer:** (B)
**Model Answer:** (B)
**Model Response:**

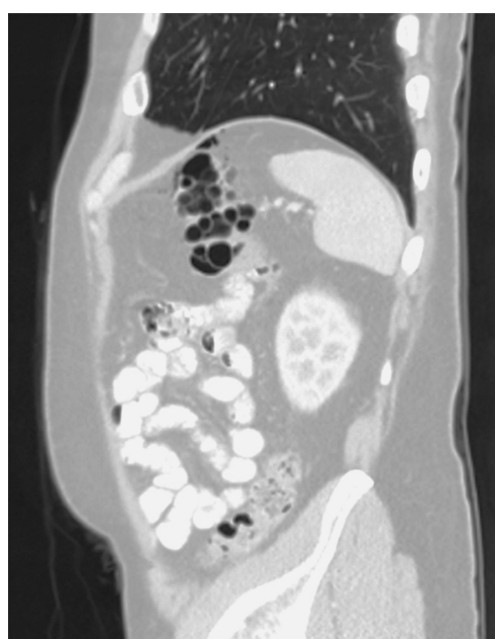

<think>The image shows a CT scan of the abdomen. The presence of gas within the intestinal wall, which is not normally seen, is a key feature of pneumatosis intestinalis. This condition is characterized by the presence of gas within the intestinal wall, which can be seen as cystic or cyst-like structures on imaging. The other options do not typically present with the same imaging findings as seen in the image. </think><answer>B. Pneumatosis intestinalis.</answer>

**Analysis:**

*The model correctly recognizes pneumatosis intestinalis by identifying gas within the bowel wall on abdominal CT. Its reasoning captures the defining radiologic feature and excludes unrelated options, showing solid diagnostic understanding. However, it lacks detailed image anchoring (e.g., cystic or linear intramural lucencies separated from luminal gas) and fails to discuss differential cues or clinical severity markers. The model accurately identifies pneumatosis intestinalis based on intramural gas on CT. Its reasoning is directionally correct but could be strengthened by specifying the cystic/linear gas pattern along the bowel wall and differentiating from mimics such as hepatic cysts or intraluminal masses.*

## MedVLSynther-7B 13K output for PathVQA

**Question**: does mycobacterium avium infection in a duodenal biopsy from a patient with aids show massive intracellular macrophage infection with acid-fast organisms filamentous and pink in this acid-fast stain preparation? **Choices:** (A) Yes (B) No

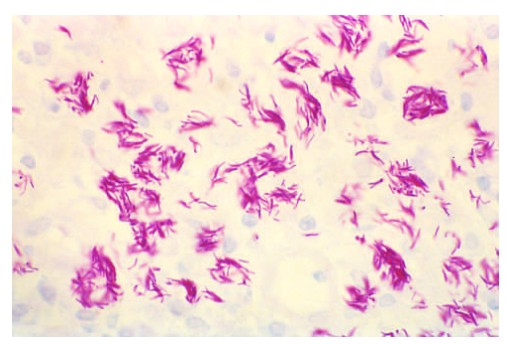

**Correct Answer:** (A)

**Model Answer:** (A)

**Model Response:**

<think>The image provided shows a high magnification view of a duodenal biopsy stained with an acid-fast stain, which is typically used to detect Mycobacterium species. The organisms appear as pink, filamentous structures within the cells, which is characteristic of Mycobacterium species. The presence of these organisms within macrophages is also consistent with a mycobacterial infection. Given the appearance of the organisms and their location within macrophages, the answer is likely to be positive.

Therefore, the answer is: </think><answer>A. Yes.</answer>

---

**Analysis:**

*The model correctly identifies that in an AIDS patient, a duodenal biopsy showing numerous pink acid-fast bacilli within macrophages on AFB stain is diagnostic of Mycobacterium avium complex (MAC) infection. Its reasoning appropriately connects the presence of acid-fast organisms and intracellular accumulation in macrophages with the characteristic histopathologic pattern of MAC enteritis. The answer is accurate and clinically well-grounded, though the description "filamentous" is slightly imprecise since these organisms are slender rods rather than true filaments. A more complete explanation would note that these AFB-laden foamy macrophages occupy the lamina propria and are highlighted by Ziehl–Neelsen or Fite stains, confirming the diagnosis of massive intracellular mycobacterial infection.*

---

**MedVLSynther-7B 13K output for PMC-VQA**

**Question**: What does the surrounding edema suggest? **Choices:** (A) A healthy brain tissue
(B) The cause of intraparenchymal hemorrhage
(C) An underlying mass
(D) A normal imaging finding
**Correct Answer:** (C)
**Model Answer:** (C)
**Model Response:**
<think>The image shows a CT scan of the brain with annotations indicating a mass within the right ventricle and hemorrhage within the right ventricle. The term "edema" is used to describe the surrounding area, which typically refers to swelling or fluid accumulation. In the context of a brain scan, edema often suggests an underlying mass or pathology that is causing the brain tissue to swell. The presence of edema in the vicinity of a mass is a common finding and is often indicative of the mass's effect on the surrounding brain tissue. </think><answer>C. An underlying mass.</answer>

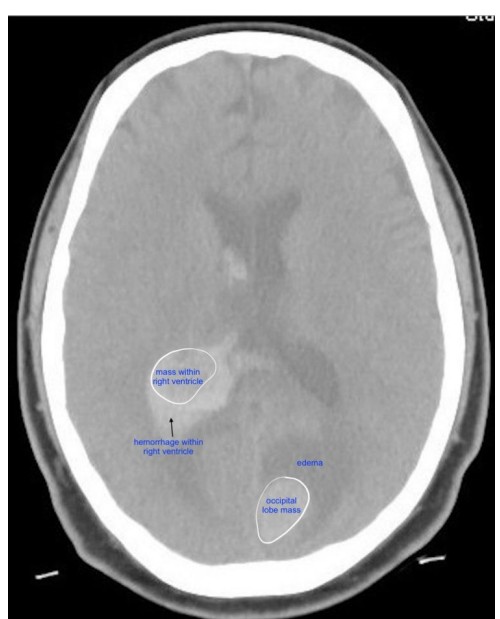

---

**Analysis:**
*The model appropriately links the presence of surrounding edema to an underlying mass, recognizing that perilesional hypodensity on non-contrast brain CT typically reflects vasogenic edema caused by tumor-induced blood–brain barrier disruption. This interpretation is anatomically and pathophysiologically sound, as such edema tracks along white matter, spares cortical ribbon, and often produces mass effect—features consistent with a neoplastic process rather than normal tissue or benign change. While the model's reasoning is correct, it could be improved by specifying the edema type (vasogenic vs. cytotoxic) and explicitly excluding other causes such as reactive changes or hemorrhage origin. Overall, the conclusion is accurate and clinically coherent: surrounding edema in this context indicates an underlying mass.*

---

**MedVLSynther-7B 13K output for SLAKE**

**Question**: Does the picture contain kidney?
**Choices:** (A) Yes
(B) No
**Correct Answer:** (B)
**Model Answer:** (B)
**Model Response:**
<think>The model correctly identifies the image as a thoracic CT slice and reasons that kidneys are absent because they lie in the abdominal region (around L1–L3). Its inference is anatomically sound, grounded in visible thoracic structures such as the heart, lungs, and ribs. The reasoning shows solid spatial and physiological understanding, though it could be strengthened by explicitly citing thoracic landmarks and noting that lower chest cuts may occasionally include the renal upper poles. </think><answer>B. No</answer>

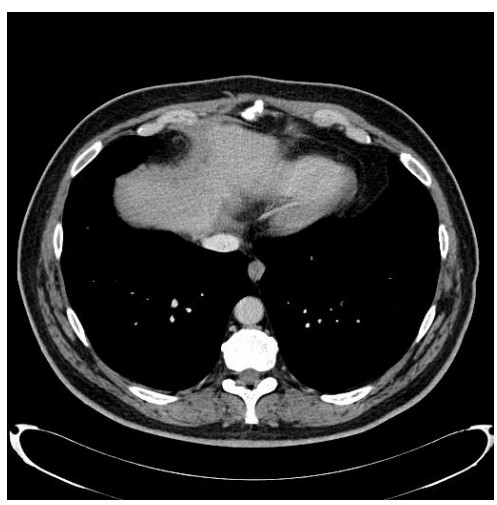

**Analysis:**
*The model correctly identifies the modality as MRI, which matches the image (a coronal brain slice with high soft-tissue contrast). It can capture clues such as soft tissue contrast and arrows indicating lesions, demonstrating its sensitivity to imaging features.*

### MedVLSynther-7B 13K output for VQA-RAD

**Question**: is there evidence of pulmonary edema? **Choices:** (A) Yes
(B) No
**Correct Answer:** (B)
**Model Answer:** (B)
**Model Response:**

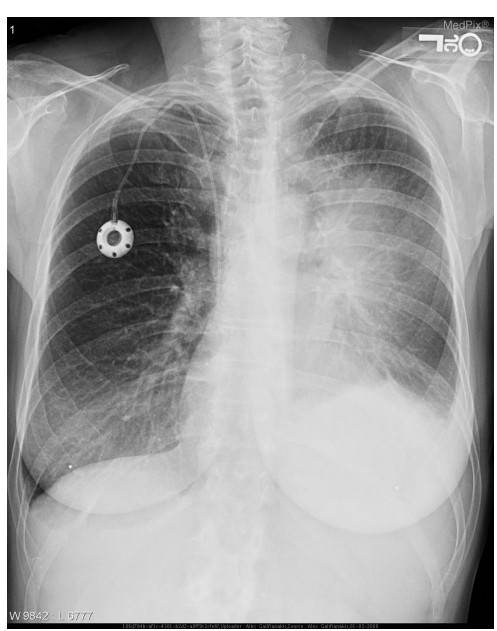

<think>To determine if there is evidence of pulmonary edema in the given chest X-ray, we need to look for certain signs. Pulmonary edema is characterized by the presence of fluid in the alveoli, which can lead to increased lung markings, blunting of the costophrenic angles, and possible areas of consolidation or hazy opacities in the lung fields.

In this X-ray, the lung fields appear relatively clear without significant opacities or increased markings that would suggest pulmonary edema. The heart size and mediastinal structures also appear within normal limits. There is no evidence of blunting of the costophrenic angles that would be indicative of pleural effusion, which is often associated with pulmonary edema.

Given these observations, there is no evidence of pulmonary edema in this image. </think><answer>B. No</answer>

---

**Analysis:**

*The model correctly concludes that there is no evidence of pulmonary edema, basing its answer on the absence of diffuse opacities, normal heart size, and clear costophrenic angles—all consistent with a non-edematous chest radiograph. Its reasoning reflects sound general understanding, but it remains somewhat superficial, lacking mention of hallmark interstitial and alveolar edema signs such as cephalization of pulmonary vessels, Kerley B lines, peribronchial cuffing, indistinct vascular margins, or perihilar "bat-wing" opacities. A stronger explanation would explicitly state that vascular markings are crisp, lung fields remain lucent without interstitial or alveolar filling, the cardiothoracic ratio is normal, and no effusions are seen. These combined findings confidently exclude pulmonary edema.*

