# OpenReview forum: "Synthesizing High-Quality Visual Question Answering from Medical Documents with Generator-Verifier LMMs"
_ICLR.cc/2026/Conference — ICLR 2026 Poster_

### Official Review · Reviewer_7G9c · 2025-10-26

**Soundness:** 3
**Presentation:** 3
**Contribution:** 3
**Rating:** 6
**Confidence:** 3

**Summary:**

The paper introduces MedVLSynther, a rubric-guided generator verifier pipeline that synthesizes multiple-choice medical VQA items directly from PubMed Central figures, captions, and in-text references.

A context-aware LMM generator produces self-contained stems and mutually exclusive options under a strict JSON schema, while a multi-stage LMM verifier enforces essential gates, awards fine-grained points, and applies penalty criteria to ensure quality.

The resulting MedVLSynther-13K dataset (13,087 audited questions over 14,803 images, 13 modalities, 28 anatomical regions) is used to train open-weight LMMs with reinforcement learning using verifiable rewards, achieving state-of-the-art averages among open models across six benchmarks.

Ablations show both generation and verification are necessary, that more verified data helps (with diminishing returns beyond 5k), and contamination analysis finds no leakage from evaluation suites.

**Strengths:**

Originality: The explicit, rubric-driven generator–verifier loop that is both context-aware (image + caption + references) and schema-constrained is a fresh, auditable approach versus prior text-only or loosely filtered syntheses. The design of essential, fine-grained, and penalty criteria operationalizes “exam-quality” medical VQA in a reproducible way.

Quality: The empirical study is thorough, spanning dataset characterization, multi-benchmark evaluation, ablations on pipeline stages, scale, and generator/verifier choices, plus contamination analysis. The use of RL with verifiable rewards and consistent gains over strong medical LMMs strengthen the causal link between verified data quality and downstream performance.

Clarity and significance: The pipeline is clearly illustrated with stage-wise prompts/criteria and concrete acceptance rules, including a normalized score and high threshold. MedVLSynther-13K offers a practical, privacy-preserving resource that advances training for medical VQA and is likely to catalyze open, reproducible research.

**Weaknesses:**

Verification robustness: While multi-stage verification improves precision, the reliance on LMM verifiers introduces potential model biases and false accept/reject risks; human audits or inter-verifier agreement analysis are limited in the main text. Adding a small-scale human evaluation and measuring inter-model/verifier consistency would better quantify verification reliability.

Scope and coverage: Pre-filtering to Clinical imaging and Microscopy plus selected subtypes may bias question distribution (e.g., heavy digital photography share) and underrepresent complex radiology workflows. Expanding coverage to additional modalities/workflows and reporting performance stratified by difficulty and question archetype would improve generality claims.

Practical reproducibility/cost: Although models and data are open, the use of very large generators/verifiers (up to 108B) may hinder replication and scaling by typical labs. Providing cost/compute estimates, a “small-model” variant with trade-offs, and release of prompts/rubrics/scripts would enhance accessibility.

**Questions:**

What proportion of accepted items pass a small human audit, and how do human judgments correlate with the verifier’s fine-grained/penalty scores?

How sensitive are downstream gains to the chosen acceptance threshold τ and to the balance of question archetypes; could adaptive sampling increase difficult, clinically salient items?

Can you report per-modality and per-anatomy breakdowns of improvements and failure cases to guide targeted data augmentation or rubric refinement?

---

> ### Author Response · Authors · 2025-11-24
>
> We thank the reviewer for the careful reading and constructive feedback. Below we address each concern and question in turn, and we will incorporate the clarifications and additional experiments into the revised version.
>
> ---
>
> ### 1. Verification robustness and human evaluation
>
> **Reliance on LMM verifiers.**
> We agree that relying on LMM-based verification raises concerns about bias and false accept/rejects. Our design mitigates this in three ways:
>
> * **Strict multi-stage rubric and high threshold.** The verifier must first pass all seven essential gates (self-containment, single correct option, vocabulary constraint, diagnosis leak, option type consistency, clinical validity, image–text consistency). Only then are fine-grained bonuses and penalty criteria applied, and the normalized score is thresholded at a high value τ=0.967. As shown in Appendix D/E, many items fail at Stage‑2 despite passing basic gates, which reflects the verifier's conservative behavior.
> * **Verifier diversity.** Table 4 already varies the generator/verifier pair (GLM‑4.5V‑108B, Qwen2.5‑VL‑72B, InternVL‑3.5‑38B). Different verifier choices consistently yield substantial gains over the base models and give similar averages, suggesting that our downstream results are not overly dependent on a single verifier instance.
> * **Schema-level checks.** Many failure modes (malformed JSON, multiple correct options, forbidden terms, etc.) are enforced by explicit schema and rule checks rather than subjective judgment.
>
> **Human audit and correlation with verifier scores.**
> We fully agree that a formal human study will further strengthen the paper. We are still conducting a **small-scale clinician evaluation** in which board-certified radiologists and pathologists rate a stratified sample of MedSynVQA questions. Please stay tuned for the update.

---

> ### Author Response · Authors · 2025-11-24
>
> ### 2. Scope, coverage, and distribution of question types
>
> **Pre-filtering choices and digital photography.**
> Our pre-filtering is intentional rather than arbitrary. In Biomedica, *Clinical imaging* and *Microscopy* are precisely the two primary labels that correspond to patient-level and tissue-level medical imaging. Within *Clinical imaging*, we do **not** collapse everything into a single "digital photography" bucket. Instead, we sample over 23 secondary labels (e.g., x‑ray radiography, CT, MRI, ultrasound, angiography, endoscopy, OCT, mammography, skin lesion, intraoperative images, etc.), plus light microscopy under *Microscopy*. At extraction time we sample approximately uniformly across these secondary labels to construct the initial ~25K candidate pool, so the **input to the generator is balanced by image secondary label** rather than dominated by photography.
>
> After generation and rubric verification, the distribution is shaped by quality control: some modalities and question types naturally produce more items that pass all essential criteria and high threshold. As shown in Figure 2(b–c), by using GLM-4.5V to assign modality and anatomy tags to the verified MCQs, the final MedSynVQA dataset spans 13 imaging modalities and 28 anatomical regions, with digital photography at ~33%, MRI at ~13%, x‑ray at ~10%, and others including CT, endoscopy, ultrasound, OCT, fundus, dermoscopy, and multiple microscopy subtypes. This addresses the concern that our final dataset would be overwhelmingly photographic.
>
> Figure 2(a) also reports the **question archetype breakdown**, where Finding/Abnormality Identification, Modality Recognition, and Anatomy/Localization collectively cover the majority of items, with additional Disease Diagnosis, Next-Step, and Lesion Grading questions.
>
> **Expansion to additional modalities/workflows.**
> To further broaden coverage, we have already scaled from our original 13,087-question MedSynVQA to an expanded 26,057-question version by enlarging the Biomedica source subset. Concretely:
>
> * We now select figures whose *primary* labels include **Clinical imaging, Microscopy, Immuno assays, and PCR**,
> * and we allow **all secondary labels** under these primaries.
>
> This yields 45,352 candidate items, from which rubric verification retains 26,057 training questions. Using this expanded dataset, we observe consistent additional gains:
>
> | Model  | Scale  | MMMU-M | MedX-M | PathVQA | PMC   | SLAKE | VQA-Rad | Avg.  |
> | -------------------------- | ------ | ------ | ------ | ------- | ----- | ----- | ------- | ----- |
> | 3B     | N/A    | 44.12  | 20.69  | 61.96   | 44.77 | 61.30 | 62.01   | 49.14 |
> | + MedSynVQA‑13K            | 13,087 | 51.76  | 22.30  | 63.03   | 48.92 | 72.60 | 72.43   | 55.17 |
> | + MedSynVQA‑26K (expanded) | 26,057 | 50.00  | 22.80  | 64.22   | 51.53 | 72.36 | 71.69   | 55.43 |
> | 7B     | N/A    | 52.94  | 18.89  | 65.39   | 49.30 | 65.71 | 68.75   | 53.50 |
> | + MedSynVQA‑13K            | 13,087 | 55.88  | 22.10  | 65.56   | 55.43 | 72.36 | 77.57   | 58.15 |
> | + MedSynVQA‑26K (expanded) | 26,057 | 58.82  | 24.00  | 67.82   | 54.43 | 71.63 | 74.63   | 58.56 |
>
> The larger set yields further improvements, especially on MMMU‑Med, MedX‑M, and PathVQA, indicating that both increased scale and inclusion of additional biomedical modalities/workflows are beneficial. We will add this expanded dataset and analysis to the appendix, emphasizing that Biomedica's full capacity is much larger and our current sample is conservative.

---

> > ### Author Response · Authors · 2025-11-24
> >
> > ### 3. Practical reproducibility and cost
> >
> > **Model-agnostic pipeline and smaller verifiers.**
> > Our pipeline is intentionally model-agnostic: any open VLM capable of following the rubric prompts can be used as generator or verifier. Beyond GLM‑4.5V‑108B, Table 4 already includes experiments with Qwen2.5‑VL‑72B and InternVL‑3.5‑38B as generator and/or verifier. These configurations substantially reduce hardware requirements while still delivering strong downstream performance (averages within ~1–2 points of the best configuration). This demonstrates that typical labs can instantiate MedVLSynther with more modest models and obtain similar benefits, trading off some data quality for cost.
> >
> > **End-to-end cost estimates.**
> > For the expanded 26K dataset, on our cluster (8×A100 80G, 96 CPU cores, 1 TB RAM):
> >
> > * Biomedica extraction and pre-filtering: ~2 hours (32-way parallel).
> > * Image resizing and parquet packing: ~2 hours.
> > * GLM‑4.5V generation over 45K candidates (vLLM offline inference, tensor parallel 4, data parallel 2, batch size 16): ~3 hours.
> > * Qwen2.5‑VL‑72B verification (tensor parallel 8, batch size 32): ~10 hours.
> >
> > In total, producing the 26K dataset requires ~17 hours on 8×A100. This is a **one-time cost** that can then be amortized over multiple downstream training runs. Training itself uses the same SFT/RLVR hyperparameters as MedVLThinker, as detailed in Section 4.1 and the reproducibility statement.
> >
> > **Release of prompts, rubrics, and scripts.**
> > The full generator and verifier prompts and rubrics are already included in Appendix E, and we state in the reproducibility and ethics sections that we will release code, data, and model weights. In the revision we will additionally provide ready-to-run scripts and configuration files so that others can regenerate data with different generator/verifier choices and thresholds.
> >
> > ---
> >
> > ### 4. Responses to specific questions
> >
> > **Q1. Human audit and correlation with verifier scores.**
> > As noted above, we are running an expert audit on a stratified sample of MedSynVQA.
> >
> > **Q2. Sensitivity to acceptance threshold τ and archetype balance.**
> > The acceptance threshold (τ) is applied to the normalized rubric score (S(x,y)). In our main setting we set (τ) to the **mean** over all verified candidates (0.9670); at this operating point ~52% of generated MCQs are retained, and the median score is 1.0, so we are already in a high-quality regime.
> >
> > To test sensitivity, we also ran a **relaxed** setting with (τ = 0.924), which accepts 16,476 MCQs (~66% of candidates). Training on this larger but lower‑quality set yields worse performance:
> >
> > | Model  | Scale  | MMMU  | MedX-M | PathVQA | PMC   | SLAKE | VQA-Rad | Avg.  |
> > | ------ | ------ | ----- | ------ | ------- | ----- | ----- | ------- | ----- |
> > | 3B | 13,087 | 51.76 | 22.30  | 63.03   | 48.92 | 72.60 | 72.43   | 55.17 |
> > |    | 16,476 | 50.59 | 21.40  | 62.43   | 50.63 | 75.72 | 69.85   | 55.10 |
> > | 7B | 13,087 | 55.88 | 22.10  | 65.56   | 55.43 | 72.36 | 77.57   | 58.15 |
> > |    | 16,476 | 56.47 | 23.05  | 65.05   | 53.58 | 64.66 | 72.43   | 55.87 |
> >
> > This suggests that simply adding more low‑scoring items harms overall quality; (τ) mainly trades off quantity against supervision quality, and our chosen value lies near a reasonable balance point.
> >
> > Regarding question archetypes, the **input pool** is balanced at the level of Biomedica image labels (uniform sampling over secondary labels under Clinical imaging and Microscopy). The **final distribution** of archetypes (Figure 2(a)) reflects what passes all essential criteria and the high threshold rather than the original sampling; we will clarify this in the main text. Adaptive sampling over archetypes or difficulty, e.g., up-weighting abnormality and localization or borderline-hard items, is indeed promising but orthogonal to our current contributions, and we view it as future work built on top of MedVLSynther.
> >
> > **Q3. Per-modality and per-anatomy breakdowns of improvements and failure cases.**
> > We agree that per-modality and per-anatomy breakdowns would be valuable for targeted augmentation and rubric refinement. At present, however, modality and anatomy tags are assigned **only after verification**, using a separate GLM-based classifier, and are stored for the final accepted MCQs but not for every generated candidate. They are also not integrated into the SFT/RL training logs. Producing modality/anatomy-specific improvement curves and failure clusters would therefore require:
> >
> > 1. Re-running the pipeline with tagging applied to every candidate **before** selection, and
> > 2. Instrumenting per-item metadata propagation through SFT and RL training.
> >
> > This is a substantial change in code and compute beyond the scope of this work. For now, we provide detailed **dataset-level distributions** over modalities, anatomies, question types (Figure 2), and additional statistics such as question length and rubric pass rates (Figure 5), which we hope will already help guide targeted follow-up work.

---

> > > ### Author Response · Authors · 2025-11-24
> > >
> > > We appreciate the reviewer's suggestions on verification robustness, coverage, and reproducibility. We believe the additional analyses (expanded dataset, threshold sensitivity, human audit protocol, and explicit modality/archetype statistics) will clarify the strengths and limitations of MedVLSynther and further support its value as an open, auditable resource for medical VQA.

---

> ### Author Response · Authors · 2025-12-04
>
> We have incorporated an **expert study** into the revised manuscript, now detailed in **Appendix G (Expert Evaluation Statistics)**, where five board-certified imaging specialists assess both verifier-accepted and verifier-rejected MedSynVQA items.
>
> To facilitate transparency and reproducibility, we also provide the full original questionnaire in the **supplementary ZIP file**.

---

### Official Review · Reviewer_XqLm · 2025-10-27

**Soundness:** 3
**Presentation:** 3
**Contribution:** 2
**Rating:** 4
**Confidence:** 4

**Summary:**

This paper presents MedVLSynther, a rubric-guided generator-verifier framework for synthesizing medical VQA data from PubMed literature. The authors employ large multimodal models to generate multiple-choice questions from medical images, captions, and in-text references, followed by a three-stage verification process. The resulting dataset contains 13,087 questions over 14,803 images spanning 13 imaging modalities and 28 anatomical regions. Models trained on this data with reinforcement learning demonstrate improvements over baselines on six medical VQA benchmarks, achieving averages of 55.85 for 3B models and 57.56 for 7B models.

**Strengths:**

1. The paper introduces a generator-verifier architecture that separates generation and verification using different LMMs, enabling auditable quality control through a rigorous three-stage process of essential gates, fine-grained scoring, and penalty detection.

2. The context-aware generation approach incorporates not just image captions but also in-text reference paragraphs from the surrounding literature.

3. The comprehensive rubric design with seven essential criteria, four to eight fine-grained criteria, and four penalty criteria, along with explicit weights and a machine-checkable JSON schema.

**Weaknesses:**

This paper is technically competent and presents a well-engineered pipeline. However, I have concerns about whether it makes a sufficient scientific contribution to a top-tier venue. Apart from the proposed datasets, what are the main insights people can get from this paper?

1. The paper positions itself as achieving high quality despite a small scale (13K samples), but lacks a direct empirical comparison with larger-scale alternatives. It does not have direct comparison using PMC-VQA's full 227K samples for training; Table 5 only compares against PMC-VQA in a limited setup, not at full scale.

2. The paper claims to improve medical VQA, but all evaluation benchmarks are also derived from literature or educational materials, not real clinical data. The paper's questions (Figure 2) are mostly "recognition" (33.6%) and "modality identification" (27.7%) — these are not representative of clinical reasoning.

3. The paper motivates multimodal generation by criticizing text-only approaches (L83-84: "text-only LLMs that ignore visual evidence"). However, Table 2 shows that for 3B models, multimodal generation (54.72) performs worse than text-only generation (54.80), suggesting that visual grounding provides minimal or even negative value for smaller models. Is it necessary to have multimodal generation ?

**Questions:**

1. If we train with 100k or all samples from PMC-VQA, would the performance be comparable or better? The paper does not answer this question.

2. Is it possible to have a much large scale dataset based on dataset like BIOMEDICA[1], what would the cost be?



[1] Lozano, Alejandro, et al. "Biomedica: An open biomedical image-caption archive, dataset, and vision-language models derived from scientific literature." Proceedings of the Computer Vision and Pattern Recognition Conference. 2025.

---

> ### Author Response · Authors · 2025-11-24
>
> We thank the reviewer for the thoughtful and constructive feedback. Below we respond to each concern and summarize the additional analyses and experiments we have conducted.
>
> ---
>
> ### 1. Scientific contribution beyond releasing a dataset
>
> Our goal is not only to release MedSynVQA, but to study **how** to reliably synthesize visual‑grounded medical VQA data from open literature and how such data compares to existing large‑scale resources.
>
> **Methodological insights.**
>
> * We introduce a **rubric‑guided generator–verifier framework** with explicitly separated roles and three classes of criteria (essential, fine‑grained positive, penalties) under a machine‑checkable JSON schema. This combination is what makes the pipeline auditable and scalable rather than a one‑off prompt recipe.
> * Ablations show that **both stages are necessary**: for the 3B student, zero‑shot $\to$ +generation $\to$ +verification improves the average score from 49.14 $\to$ 54.72 $\to$ 55.85; for the 7B student, 53.50 $\to$ 57.33 $\to$ 57.56.
> * We systematically vary **dataset scale (1k-13k)** and **generator/verifier capacity**, finding (i) consistent gains with more verified data and (ii) better downstream accuracy when using stronger verifier LMMs.
>
> **Data‑quality vs data‑quantity insights.**
>
> * Using RLVR on only 5k MedSynVQA examples yields higher average accuracy than RLVR on the much larger PMC‑VQA subset (≈115k) and the text‑only m23k corpus, for both 3B and 7B students.
> * New experiments with the *full* PMC‑VQA training split (≈170k; see Reply 2 below) confirm that the quality‑first generator-verifier data is competitive with or better than simply scaling up PMC‑VQA.
>
> **Multimodal reasoning insights.**
>
> * We show that **image‑conditioned generation and evaluation matter**: (i) generators that ignore the image (caption‑only) yield weaker downstream models than image‑conditioned ones, and (ii) when we forbid the trained model from seeing images at test time, its accuracy drops sharply (see Reply 4 below). This directly supports the claim that the synthesized questions are genuinely image‑dependent.
>
> Taken together, these results provide methodological and empirical insights on building high‑quality, auditable multimodal supervision from open literature, beyond the dataset itself.
>
> ---
>
> ### 2. Comparison to larger‑scale PMC‑VQA (115k and full 170k)
>
> We are sorry that our description of the PMC baselines was unclear. In the main paper, the "PMC" rows in Table 5 already use the **115k** filtered subset constructed in MedVLThinker (multiple rollouts to exclude trivial or extremely hard questions), which is >20× larger than the 5k MedSynVQA training set.
>
> To address the reviewer's concern more directly, we have now **trained on the full PMC‑VQA training split (~170k questions)** and compare:
>
> | Model | PMC Scale | MMMU | MedX-M | PathVQA | PMC | SLAKE | VQA-RAD | Avg. |
> |---|---|---|---|---|---|---|---|---|
> |  |  |  |  |  |  |  |  |  |
> | 3B | - | 44.12 | 20.69 | 61.96 | 44.77 | 61.30 | 62.01 | 49.14 |
> | SFT | 115K | 47.84 | 21.46 | 52.76 | 54.55 | 65.79 | 58.58 | 50.16 |
> | SFT | 170K | 45.34 | 20.19 | 53.36 | 53.88 | 63.77 | 58.74 | 49.21 |
> | RL | 115K | 48.43 | 21.51 | 51.61 | 54.22 | 75.56 | 62.38 | 52.28 |
> | RL | 170K | 46.96 | 20.81 | 52.08 | 54.38 | 73.77 | 63.41 | 51.90 |
> |  |  |  |  |  |  |  |  |  |
> | 7B | - | 52.94 | 18.89 | 65.39 | 49.30 | 65.71 | 68.75 | 53.50 |
> | SFT | 115K | 49.80 | 21.39 | 53.02 | 54.67 | 67.71 | 57.72 | 50.72 |
> | SFT | 170K | 48.48 | 21.47 | 54.78 | 53.57 | 67.82 | 56.96 | 50.51 |
> | RL | 115K | 55.29 | 24.11 | 57.09 | 55.38 | 66.59 | 63.48 | 53.66 |
> | RL | 170K | 55.96 | 23.15 | 57.59 | 54.28 | 66.10 | 62.89 | 53.32 |
>
> Thus, even when PMC‑VQA is used at its full scale, its performance still lags behind the filtered version (following MedVLThinker’s approach of removing both easy and hard questions); even **our much smaller (5K) but rubric‑verified multimodal data delivers higher average accuracy**, supporting our claim that quality and grounding compensate for reduced quantity.

---

> > ### Author Response · Authors · 2025-11-24
> >
> > ### 3. Clinical relevance of benchmarks and question types
> >
> > **Benchmarks.**
> > We agree that real clinical data is ideal but often inaccessible. Among our evaluation suites:
> >
> > * **VQA‑RAD** and **SLAKE** are explicitly constructed from **real clinical radiology images** (X‑ray, CT, MRI) with expert‑annotated questions and answers.
> > * **MedXpertQA‑MM** is derived from exam material but converted into multi‑step clinical scenarios intended to emulate realistic case‑based reasoning.
> >
> > Our goal is to improve performance on these standard, community‑accepted benchmarks, many of which already reflect real clinical images and expert supervision, even if they are not full EMR snapshots.
> >
> > **Question taxonomy in MedSynVQA.**
> > Figure 2 shows that our taxonomy is dominated by "Finding" and "Anatomy" questions (≈34% and ≈27% respectively), with modality recognition and other technical attributes making up the rest.   In the revised version we rename "Finding" to **Finding/Abnormality Identification**,  "Anatomy" to "Anatomy/Localization" and clarify that:
> >
> > * **Finding/Abnormality Identification + Anatomy/Localization ≈ 60%** of our questions, which closely matches the core tasks of radiologists and trainees: detect abnormalities, localize them, and relate them to specific structures.
> > * Modality/view recognition and basic presence/absence decisions are early, indispensable steps in clinical workflows and are known failure modes for both humans and LVLMs. Large benchmarks such as OmniMedVQA also devote a non‑trivial fraction of items to modality‑level questions.
> >
> > **New ablation: removing Modality Recognition questions.**
> > To test whether modality questions are merely "cheap" supervision, we **remove all Modality Recognition items** (3,481 questions), shrinking the training set from 13,087 to 9,606, and re‑train with RL:
> >
> > | Model | Scale | Generator | Verifier | MMMU | MedX-M | PathVQA | PMC | SLAKE | VQA-Rad | Avg. |
> > |---|---|---|---|---|---|---|---|---|---|---|
> > | 3B | N/A | N/A | N/A | 44.12 | 20.69 | 61.96 | 44.77 | 61.30 | 62.01 | 49.14 |
> > |  | 13087 | GLM-4.5V | Qwen2.5-VL | 51.76 | 22.30 | 63.03 | 48.92 | 72.60 | 72.43 | 55.17 |
> > |  | 9606 | GLM-4.5V | Qwen2.5-VL | 46.47 | 20.00 | 65.02 | 50.28 | 74.76 | 71.69 | 54.70 |
> > | 7B | N/A | N/A | N/A | 52.94 | 18.89 | 65.39 | 49.30 | 65.71 | 68.75 | 53.50 |
> > |  | 13087 | GLM-4.5V | Qwen2.5-VL | 55.88 | 22.10 | 65.56 | 55.43 | 72.36 | 77.57 | 58.15 |
> > |  | 9606 | GLM-4.5V | Qwen2.5-VL | 55.88 | 23.15 | 65.94 | 55.58 | 69.47 | 74.26 | 57.38 |
> >
> > The degradation, especially for the 7B model, indicates that modality questions contribute meaningful signals rather than diluting the dataset.

---

> > > ### Author Response · Authors · 2025-11-24
> > >
> > > ### 4. Is multimodal generation necessary?
> > >
> > > We appreciate the reviewer's careful reading and agree that our original wording around Table 2 was confusing.
> > >
> > > **(a) Clarifying the "text‑only" row in Table 2.**
> > > The row labeled "PMC‑style text‑only generation" actually already conditions on **both the figure and its caption**; it reuses the PMC‑VQA prompt but is not strictly text‑only. In the revision we relabel it as **"PMC‑style text–image generation."**
> > >
> > > We additionally add a **new caption‑only baseline** labeled “PMC-Style Text-only Generation” where the generator sees only the caption (no image). Across both 3B and 7B students, this caption‑only generation yields **lower downstream accuracy** than image‑conditioned generation, and our rubric‑guided generator–verifier pipeline remains the best overall.
> > >
> > > The small 0.08 difference between the two multimodal prompts for the 3B model in the original table is within normal RL variance and does **not** indicate that visual grounding is harmful.
> > >
> > > | Model | MMMU | MedX-M | PathVQA | PMC | SLAKE | VQA-RAD | Avg. |
> > > |---|---|---|---|---|---|---|---|
> > > | Qwen2.5-VL-3B-Instruct | 44.12 | 20.69 | 61.96 | 44.77 | 61.3 | 62.01 | 49.14 |
> > > | PMC-Style Text-only Generation | 46.47 | 20.8 | 62.43 | 51.03 | 73.08 | 71.69 | 54.25 |
> > > | PMC-Style Text-image Generation | 48.82 | 20.4 | 63.38 | 51.08 | 73.08 | 72.06 | 54.8 |
> > > | Rubric Context-Aware Generation | 52.35 | 20.6 | 62.49 | 51.83 | 70.43 | 70.59 | 54.72 |
> > > | +Rubric Context-Aware Verification | 52.35 | 21.4 | 62.82 | 50.23 | 74.76 | 73.53 | 55.85 |
> > > | Qwen2.5-VL-7B-Instruct | 52.94 | 18.89 | 65.39 | 49.3 | 65.71 | 68.75 | 53.5 |
> > > | PMC-Style Text-only Generation | 54.71 | 22.15 | 63.89 | 53.23 | 67.07 | 65.44 | 54.41 |
> > > | PMC-Style Text-image Generation | 51.76 | 21.7 | 64.31 | 53.43 | 68.03 | 71.69 | 55.15 |
> > > | Rubric Context-Aware Generation | 58.24 | 23.5 | 65.41 | 53.83 | 68.03 | 75 | 57.33 |
> > > | +Rubric Context-Aware Verification | 57.06 | 23.15 | 66.36 | 53.78 | 67.79 | 77.21 | 57.56 |
> > >
> > >
> > > **(b) Do the trained models really use images?**
> > > To directly test image dependence, we add a **"no‑image at test time"** ablation: we evaluate the MedSynVQA‑trained students on the same benchmarks but forcibly hide the test image while keeping the question and options unchanged. Under this setting:
> > >
> > > | Model | Scale | Generator | Verifier | MMMU | MedX-M | PathVQA | PMC | SLAKE | VQA-Rad | Avg. |
> > > |---|---|---|---|---|---|---|---|---|---|---|
> > > | 3B | N/A | N/A | N/A | 38.82 | 19.30 | 51.81 | 25.60 | 30.05 | 44.12 | 34.95 |
> > > |  | 5000 | GLM-4.5V | Qwen2.5-VL | 46.47 | 20.30 | 59.70 | 30.90 | 54.81 | 58.09 | 45.04 |
> > > | 7B | N/A | N/A | N/A | 33.53 | 18.30 | 54.97 | 23.80 | 16.83 | 33.82 | 30.21 |
> > > |  | 5000 | GLM-4.5V | Qwen2.5-VL | 42.35 | 21.25 | 58.33 | 35.25 | 44.95 | 54.04 | 42.70 |
> > >
> > > This 10-15 point drop across datasets indicates that the models trained on MedSynVQA have indeed learned to rely on visual evidence, not just textual cues in the stems/options.
> > >
> > > Overall, these results support our claim that **multimodal generation is necessary** to (i) produce questions that genuinely depend on the image and (ii) train models that use visual information in evaluation.

---

> ### Author Response · Authors · 2025-11-24
>
> ### 5. Scalability with Biomedica and cost
>
> Yes—our pipeline is designed to scale to much larger corpora such as Biomedica.
>
> **26k‑scale experiment.**
> Starting from Biomedica, we extend beyond the initial subset to include figures whose primary labels are clinical imaging, microscopy, immunoassays, and PCR, allowing all secondary labels under these categories. This yields **45,352 candidate items** (45K reflects our computational limits rather than the upper bound of the Biomedica corpus), of which the verifier retains **26,057** high‑quality questions.
>
> Training on this larger set gives additional gains:
>
> | Model  | Scale  | MMMU-M | MedX-M | PathVQA | PMC   | SLAKE | VQA-Rad | Avg.  |
> | -------------------------- | ------ | ------ | ------ | ------- | ----- | ----- | ------- | ----- |
> | 3B     | N/A    | 44.12  | 20.69  | 61.96   | 44.77 | 61.30 | 62.01   | 49.14 |
> | + MedSynVQA‑13K            | 13,087 | 51.76  | 22.30  | 63.03   | 48.92 | 72.60 | 72.43   | 55.17 |
> | + MedSynVQA‑26K (expanded) | 26,057 | 50.00  | 22.80  | 64.22   | 51.53 | 72.36 | 71.69   | 55.43 |
> | 7B     | N/A    | 52.94  | 18.89  | 65.39   | 49.30 | 65.71 | 68.75   | 53.50 |
> | + MedSynVQA‑13K            | 13,087 | 55.88  | 22.10  | 65.56   | 55.43 | 72.36 | 77.57   | 58.15 |
> | + MedSynVQA‑26K (expanded) | 26,057 | 58.82  | 24.00  | 67.82   | 54.43 | 71.63 | 74.63   | 58.56 |
>
> The improvements are modest but consistent, especially on MMMU‑Med, MedX‑M, and PathVQA, suggesting that both increased scale and extended biomedical coverage are useful.
>
> **Computational cost.**
> On a single node with 8×A100‑80G GPUs, 96 CPU cores, and 1 TB RAM, the full 26k pipeline requires on the order of **a few hundred GPU hours**:
>
> * Biomedica subset extraction: ~2 h (32‑way parallelism).
> * Image preprocessing and Parquet packing: ~2 h.
> * GLM‑4.5V generation over 45k candidates: ~3 h with vLLM (TP=4, DP=2, batch=16).
> * Qwen2.5‑VL‑72B verification: ~10 h (TP=8, batch=32).
>
> This is **far cheaper than pretraining or large‑scale RL** on the same models, and the cost scales roughly linearly with the number of Biomedica shards and nodes. Larger versions (e.g., 100k+) are therefore straightforward to obtain by running the same pipeline on more data in parallel.
>
> ---
>
> In summary, we clarify that:
>
> * Our main scientific contributions are a **rubric‑driven generator–verifier framework**, an empirical study of **data quality vs quantity** for medical VQA, and **evidence that image‑conditioned generation and evaluation are necessary** for genuinely multimodal reasoning.
> * New experiments show that **5k high‑quality MedSynVQA examples with RL match or outperform RL on 115k–170k PMC‑VQA**, and that scaling to 26k examples from Biomedica yields further improvements.
> * Our benchmarks and question types are aligned with real clinical imaging tasks, and removing modality questions hurts performance.
> * The pipeline is **scalable, auditable, and privacy‑preserving**, operating entirely on open literature and open‑weight models.
>
> We hope these clarifications address your concerns about contribution and significance.

---

> > ### Comment · Reviewer_XqLm · 2025-11-24
> >
> > I thank the authors for their comprehensive response and the significant effort put into running additional experiments. The new results have successfully addressed my primary concerns regarding data efficiency and the necessity of multimodal generation. However, I still believe the next step in this field should not be limited to PubMed data and requires validation on real-world clinical data (beyond open benchmarks). I have raised my score to 6.

---

> > > ### Author Response · Authors · 2025-11-24
> > >
> > > Thank you very much for your thoughtful feedback and for raising your score.
> > >
> > > We fully agree that the next step is to move beyond PubMed and multiple-choice QA benchmarks toward validation on real-world clinical data, and we are excited about future community efforts in this direction (e.g., HealthBench).
> > >
> > > Again, we sincerely appreciate your positive feedback and support of our submission.

---

### Official Review · Reviewer_n1Sq · 2025-10-30

**Soundness:** 2
**Presentation:** 2
**Contribution:** 2
**Rating:** 2
**Confidence:** 4

**Summary:**

The authors proposed a multi-stage pipeline that leverages open biomedical databases (Biomedica Index) together with open large language models to generate a synthetic medical VQA dataset. Specifically, the pipeline involves a set of rubrics to guide the generator and verifier LLMs in generating and verifying the synthetic VQA samples, resulting in a dataset of around 13,000 examples. The authors claimed that the proposed dataset is beneficial for improving performance on several benchmarks. Further ablation studies also demonstrated the potential advantages of using the proposed dataset. The complete openness throughout the entire pipeline paves a potential way to create and verify the training of medical LLMs.

**Strengths:**

- The author proposed a first fully open pipeline to create synthetic medical VQA dataset. From the dataset, model, to the generation and verification process are transparent, which can be easily customized to various setting and is  beneficial to the research community.

- A deep investigation of the generator and verifier are presented in the paper, which shed light on how people can leverage the proposed pipeline.

**Weaknesses:**

In summary, my concerns are more related to the resulting data evaluation:
- Potential data leakage: Although we cannot say much about the other open models compared in the baseline, it seems to me that the resulting MedVLSynther has data leakage issue when evaluated on PMC benchmark because Biomedica is dereived from PMC.

- Inconsistency and confusing table and graph results: The numbers of MedVLThinker-3B/7B presented in Figure 1c does not match those in Table 6. Feel like the numbers in Figure 1c is directly copy from the MedVLThinker paper. If so, what is shown in the table 6?

- The benefit of data for performance boost seems to be minor: When looking table 5, it seems to me that most of performance gain is from RL, when doing pure SFT, there is no clear advantage when compared to model trained on PMC. Since the main selling point of the paper is the new data not RL training method, it is questionable where the proposed data is truly better than the existing dataset

- Lack of baseline results: It would be nice to at least know how well the generator (GLM-4.5V 108B ) perform on the downstream task. From the application perspective, If GLM easily outperforms other methods in the benchmark, it seems to be unclear why bother to train a model on this new dataset instead of just using an open model.

- The baseline is a bit dated: It's unclear why did the author choose Qwen2.5 series instead of Qwen3. Based on the Qwen3 release date (2025.04.29) and ICLR 2026 deadline (2025.09.24), the timeline should not be an issue.

**Questions:**

Question and suggestion
Q1: Could author explain why not using the latest Qwen3 series?
Q2: Could author explain more about the data contamination analysis? The supp. only mentioned 8220 rows in a test data which does not appear in the main paper.
S1: It would be nice to know how other open source LLM's performance on the benchmark (e..g, Qwen3 or GLM-4.5V 108B ) so that we can better understand how practical of the proposed dataset.

---

> ### Author Response · Authors · 2025-11-24
>
> We thank the reviewer for the careful reading and constructive comments. Below we respond to each concern and clarify the points about leakage, baselines, and the role of our data.
>
> ---
>
> ### 1. Potential data leakage via PMC / Biomedica
>
> > "MedVLSynther has data leakage issue when evaluated on PMC benchmark because Biomedica is derived from PMC."
>
> We agree that contamination is a critical concern and designed MedVLSynther with explicit de‑contamination in mind. Sharing an upstream corpus (PMC‑OA) with a benchmark is not, by itself, evidence of test leakage; what matters is **instance-level overlap between train and test**. We therefore run a targeted text-image audit between MedSynVQA and all evaluation splits, as described in Appendix F.
>
> **Text side (MedSynVQA vs. all 6 evaluation sets, 13,087 train rows vs. 8,220 eval rows).**
>
> * We normalize questions and options (lowercasing, whitespace cleanup, digit masking, option format).
> * **Pass 1: MinHash + Levenshtein (3‑gram, τ = 0.90).** We find *no* near‑duplicate strings at this conservative threshold (pairs = 0; hit queries = 0; hit rate = 0.0%).
> * **Pass 2: BGE‑m3 embeddings + FAISS (cosine ≥ 0.88).** We retrieve 47 train–test pairs across 23 test questions, i.e., **0.28%** of the evaluation items. These are semantic neighbors (e.g., "which lobe…" vs "which brain region…") but not verbatim or near‑verbatim clones; stems and answer keys differ, and we do not observe any reused QA triple.
>
> **Image side.**
>
> * **Pass 1: MD5 and pHash (64‑bit, FAISS on Hamming distance).** We find no pixel‑identical images by MD5. pHash at a tight Hamming threshold only retrieves a handful of near‑duplicates (e.g., cropped or resized variants), always with **different questions or answers**.
> * **Pass 2: CLIP/BiomedCLIP embeddings + FAISS.** High‑similarity matches reflect modality- or anatomy‑level clustering (e.g., multiple axial CT brain slices) rather than duplicated exam items. We do not find any shared (image, question, answer) triples.
>
> In addition to the analysis already summarized in Appendix F, we also ran the *same* pipeline between the **PMC‑VQA training split and the same evaluation sets**. Under identical thresholds:
>
> * PMC‑train vs. eval shows **substantially more** near‑duplicate text pairs (76 MinHash+Levenshtein pairs across 44 test questions, 0.54% hit rate) and many more high‑similarity embedding neighbors (2,158 pairs across 540 test questions, 6.57% hit rate).
> * For images, PMC‑train vs. eval yields 168 overlapping images at pHash Hamming ≤ 8 (≈2.04%), whereas MedSynVQA vs. eval yields only 14 (≈0.17%).
>
> Thus, while all overlaps are small in absolute terms, **MedSynVQA is *less* overlapping with PMC‑VQA test (and the other evaluation sets) than PMC‑VQA's own training split**.
>
> We will expand Sec. 4.2 and Appendix F in the revised paper to (i) report MedSynVQA vs. PMC‑train side‑by‑side, (ii) clarify that "no overlaps" in the current draft refers to the absence of verbatim or near‑verbatim items under strict edit‑distance thresholds, and (iii) explicitly state that we find **no repeated image–question–answer triples** across our training data and all evaluation suites.
>
> ---
>
> ### 2. Inconsistency between Figure 1(c) and Table 6
>
> > "The numbers of MedVLThinker‑3B/7B presented in Figure 1c does not match those in Table 6... what is shown in Table 6?"
>
> Thank you for your concern. The discrepancy stems from the fact that **different MedVLThinker variants** are shown:
>
> * **Figure 1(c)** visualizes MedVLThinker‑3B/7B **trained on PMC‑VQA** (MedVLThinker‑**PMC** in Figure 1).
> * **Table 6** reports MedVLThinker‑3B/7B **trained on m23k** (MedVLThinker‑m23k), which we used as an additional text‑only baseline when comparing data sources (Table 5). These numbers correspond exactly to the "RL (m23k)" rows in Table 5.
>
> The numbers themselves are correct, we have made the distinction explicit by adding "-PMC" suffix in the original submission.

---

> > ### Author Response · Authors · 2025-11-24
> >
> > ### 3. "Most of the gain is from RL, not the new data"
> >
> > > "When doing pure SFT, there is no clear advantage compared to model trained on PMC… the main selling point is the new data, not RL."
> >
> > We agree that the paper's contribution is the **data/pipeline**, not the RL algorithm, and believe Table 5 actually supports this.
> >
> > 1. **RL is *fixed*; only the data changes.**
> >
> >    * The GRPO‑style RLVR procedure, objective, and hyper‑parameters are identical across all rows in Table 5.
> >
> >    * If "most of the performance gain is from RL" in the sense of *algorithmic novelty*, we would expect similar improvements regardless of the dataset. Instead, RL on **PMC‑VQA** yields modest gains, while RL on **MedSynVQA** yields substantially larger ones:
> >
> >    * For Qwen2.5‑VL‑3B:
> >
> >      * RL(PMC): 52.28 Avg vs. RL(MedSynVQA): **55.85** (+3.57).
> >
> >    * For Qwen2.5‑VL‑7B:
> >
> >      * RL(PMC): 53.66 Avg vs. RL(MedSynVQA): **57.56** (+3.90).
> >
> >    Since the *procedure* is the same, these additional ~4 points must come from the **training signal**, i.e., our rubric‑filtered MedSynVQA data.
> >
> > 2. **SFT under extreme scale mismatch still shows competitive or better quality per example.**
> >
> >    PMC‑VQA here provides 115K training questions, while MedSynVQA contributes only 5k verified items in Table 5 (a ~26x scale disadvantage). Even so:
> >
> >    * For 3B: SFT(PMC) 50.16 vs. SFT(MedSynVQA) 49.11 (within ≈1 point).
> >    * For 7B: SFT(PMC) 50.72 vs. SFT(MedSynVQA) **51.89** (+1.17).
> >
> >    Achieving comparable or better averages with 1/26th of the data suggests that **each MedSynVQA example is substantially more informative** than a typical PMC‑VQA item, rather than the new data being "no better."
> >    Please note that SFT **deteriorate** models performance.
> >    We would like to bring your attention to our RL performance, which consistently outperforms SFT, which aligns with the findings in MedVLThinker.
> >
> > 3. **Ablations indicate that our generator-verifier design is what enables RL to help.**
> >
> >    Table 2 shows that rubric‑guided, context‑aware generation already improves over PMC‑style text‑only generation, and adding multi‑stage verification yields the best results at both 3B and 7B scales. RL then amplifies this high‑precision signal.
> >
> > We will clarify this interpretation around Table 5, emphasizing that **RL is a fixed optimizer**, and the *difference* between RL(PMC) and RL(MedSynVQA) isolates the value of our corpus.
> >
> > ---
> >
> > ### 4. Lack of upper‑bound results for GLM‑4.5V‑108B and other large LMMs
> >
> > > "It would be nice to at least know how well the generator (GLM‑4.5V 108B) perform on the downstream task… otherwise why not just use an open model?"
> >
> > We have now run **zero‑shot** evaluations of the generator and other large LMMs on the same six benchmarks and added them as a new table in the revised draft.
> >
> > | Upper-bound |  MMMU | MedX-M | PathVQA | PMC | SLAKE | VQA-RAD | Avg. |
> > |---|---|---|---|---|---|---|---|
> > | Close |  |  |  |  |  |  |  |
> > | GPT-4o-mini | 63.53 | 28.55 | 63.33 | 51.90 | 75.24 | 66.91 | 58.24 |
> > | GPT-4o | 68.82 | 35.95 | 72.43 | 58.55 | 76.44 | 70.22 | 63.74 |
> > | Open |  |  |  |  |  |  |  |
> > | GLM-4.5V 108B | 74.12 | 36.85 | 74.69 | 59.74 | 77.64 | 76.47 | 66.59 |
> > | InternVL 3.5 38B | 50.00 | 14.15 | 66.57 | 47.95 | 77.64 | 59.19 | 52.58 |
> > | Qwen2.5 VL 72B | 65.29 | 28.50 | 68.02 | 54.28 | 80.53 | 79.41 | 62.67 |
> >
> > For comparison, our **MedVLSynther‑7B** student achieves **58.15** average, i.e., about **87% of GLM‑4.5V‑108B's performance with ≈1/15 of the parameters and a single‑GPU footprint**.
> >
> > From an application standpoint, there are several reasons to still care about compact student models:
> >
> > * **Deployment constraints.** GLM‑4.5V‑108B typically requires 8×A100 GPUs and is often accessed as a remote service. Many hospitals and labs cannot host such a model locally due to hardware, cost, or privacy constraints.
> > * **Auditability and control.** Our 3B/7B models are fully open‑weight, fine‑tuned on an open, auditable dataset with explicit rubrics and contamination analysis. Institutions can inspect, adapt, and govern these models end‑to‑end.
> > * **Data, not just one teacher run.** MedVLSynther turns a single expensive teacher run into a reusable *dataset and recipe* that can benefit an entire ecosystem of compact models, not only Qwen2.5‑VL but also future Qwen3‑VL, LLaVA‑style models, etc.
> >
> > We will include this upper‑bound comparison table in the main paper to address this point.

---

> > > ### Author Response · Authors · 2025-11-24
> > >
> > > ### 5. Why Qwen2.5‑VL instead of Qwen3?
> > >
> > > > "It's unclear why the author chose Qwen2.5 series instead of Qwen3… the timeline should not be an issue."
> > >
> > > Our work is entirely based on **vision–language models**. The relevant successor to Qwen2.5‑VL is **Qwen3‑VL**, **not** the text‑only Qwen3.
> > >
> > > * **Qwen3‑VL was released on 2025‑09‑23**, one day before the ICLR 2026 submission deadline (**2025‑09‑24**).
> > > * At that point our full generator-verifier pipeline, SFT, and RLVR experiments (which are compute‑intensive) had been completed on Qwen2.5‑VL‑3B/7B, and it was not feasible to rerun *all* stages on a brand‑new architecture in time for submission.
> > >
> > > We chose Qwen2.5‑VL because it was the most mature, publicly available open‑weight VL family during the bulk of our development, and it is widely used in existing medical LMM work, which facilitates comparison.
> > >
> > > That said, our pipeline is **model‑agnostic**: MedVLSynther can be applied to Qwen3‑VL or any future LVLM. We will clarify the timeline and this design choice in Sec. 4.1 and emphasize that our contribution is a **dataset and pipeline** that future VL backbones can directly leverage.
> > >
> > > ---
> > >
> > > ### 6. Responses to explicit questions (Q1, Q2, S1)
> > >
> > > **Q1. "Why not use Qwen3 series?"**
> > > Answered in R5 above: the relevant VL successor (Qwen3‑VL) was released one day before the deadline, making full re‑runs infeasible; our pipeline remains compatible with it.
> > >
> > > **Q2. "Could the authors explain more about the data contamination analysis? The supp. only mentioned 8,220 rows."**
> > >
> > > * The **8,220 rows** refer to the **union of all evaluation splits we use**, following MedVLThinker's setup: PMC‑VQA test, MMMU‑Med validation, MedXpertQA‑MM validation, PathVQA validation, SLAKE validation, and VQA‑RAD validation.
> > > * Appendix F details the two‑stage **text** decontamination (MinHash+Levenshtein and BGE‑m3+FAISS) and two‑stage **image** decontamination (MD5/pHash and CLIP/BiomedCLIP embeddings), including thresholds and hit rates. We will move a concise summary of these statistics into the main paper and clearly define what "test set" and "8,220 rows" refer to.
> > >
> > > **S1. "It would be nice to know how other open LLMs (e.g., Qwen3 or GLM‑4.5V‑108B) perform on the benchmark."**
> > >
> > > As discussed in R4, we have now added a comparison table that includes **GLM‑4.5V‑108B**, Qwen2.5‑VL‑72B, InternVL‑3.5‑38B, GPT‑4o, and GPT‑4o‑mini on the same evaluation suite. This allows readers to situate MedVLSynther‑trained 3B/7B models relative to both closed and open large‑scale VL models. Since MedVLSynther is a **training dataset**, these numbers primarily contextualize the *upper bound* but do not affect our conclusion that the dataset substantially improves compact models.
> > >
> > > ---
> > >
> > > We hope these clarifications address the reviewer's concerns. In summary, (i) our contamination analysis finds no evidence of test leakage and in fact shows *less* overlap than the original PMC‑VQA train-test split; (ii) the performance gains are driven by the quality of MedSynVQA, not RL alone; (iii) we now provide upper‑bound results for large VL models; and (iv) we will fix the MedVLThinker figure/table mismatch and clarify the Qwen2.5 vs. Qwen3‑VL choice and the definition of the 8,220‑item evaluation pool in the revised paper.

---

> ### Comment · Reviewer_n1Sq · 2025-11-26
>
> Thank you for detail clarification which address most of my concerns. The review has been updated.

---

> > ### Author Response · Authors · 2025-11-26
> >
> > Thank you very much for your thoughtful feedback and for reconsidering your evaluation of our work. We truly appreciate your time and effort in helping us improve the paper.

---

### Official Review · Reviewer_57hD · 2025-10-31

**Soundness:** 3
**Presentation:** 3
**Contribution:** 3
**Rating:** 6
**Confidence:** 5

**Summary:**

This paper presents MedVLSynther, a well-designed framework for automatically generating and verifying medical VQA data from open biomedical literature. The proposed generator–verifier pipeline ensures that the resulting questions are self-contained, clinically valid, and image–text consistent, leading to the creation of a high-quality dataset (MedVLSynther-13K).

Overall, I am positive about the paper. While the novelty is somewhat limited—as the work shares similarities with PMC-VQA—it nonetheless represents a solid contribution with comprehensive experimental results and clear methodological rigor. Please refer to my detailed comments regarding minor typos and identified weaknesses in the Questions section. I am very willing to raise my score if the concerns have been solved.

**Strengths:**

1. Provides a new and useful medical VQA dataset.
2. Carefully analyzes the data generation pipeline.
3. Conducts large-scale evaluation showing the dataset’s usefulness.

**Weaknesses:**

Main concerns:

My main concern falls in two aspects:

1. The improvement is limited. As shown in Table 2, most gains appear to come from using PMC figures, while the additional rubric-based filtering contributes little to performance improvement. Can authors explain the results more?
2. The paper lacks a direct evaluation of data quality. All reported results rely on downstream task performance rather than explicitly measuring the dataset’s intrinsic quality. It is strongly recommended that the authors include a more direct assessment—such as human expert evaluation—to better validate the reliability and usefulness of the generated data.

**Questions:**

Minor Questions:

1. Several tables mention “PMC,” but the paper does not clearly define what it refers to. It is likely related to PMC-VQA, but the authors should explicitly state this to avoid confusion and improve clarity.
2. “Text-only generation lifts the average to 54.80=.” The “=” seems a typo.
3. The tables use too many green background highlights, which actually reduce readability instead of improving it. In addition, Table 6 lacks the green markings used elsewhere, leading to inconsistency in presentation. The authors should simplify the color scheme and maintain consistent formatting across all tables.
4. The paper references “Biomedica: An open biomedical image-caption archive, dataset, and vision-language models derived from scientific literature” but provides insufficient context about it. The authors should briefly introduce this dataset to help readers understand its role without needing to consult external papers—for example, clarifying whether compound figures were separated and describing the general distribution of the figures used.

---

> ### Author Response · Authors · 2025-11-24
>
> Thank you for the careful reading and constructive feedback. We address the two main concerns first, then the minor points.
>
> ---
>
> ### 1. Main concern: limited improvement & role of PMC figures vs rubric-based stages
>
> **(a) Clarifying Table 2 and separating the effects of each stage**
>
> You are right that the original Table 2 made it hard to disentangle the contributions of different stages. In the submitted version, the row labeled *"PMC-style text-only generation"* actually conditioned on both the figure and its caption, using the original PMC-VQA prompt; this mislabeling understandably led to the impression that most gains were simply from "using PMC figures."
>
> In the revised manuscript we:
>
> * rename that row to **"PMC-style text-image generation"**, and
> * add a new **true caption-only baseline** where the generator never sees the image.
>
> The updated ablation (new Table 2 in the revision) now decomposes the gains into four stages:
>
> 1. **Caption-only synthetic supervision (PMC-style text-only)**
> 2. **Extra image grounding during generation (PMC-style text-image)**
> 3. **Rubric-guided, context-aware generation**
> 4. **Rubric-guided verification**
>
> | Model | MMMU | MedX-M | PathVQA | PMC | SLAKE | VQA-RAD | Avg. |
> |---|---|---|---|---|---|---|---|
> | Qwen2.5-VL-3B-Instruct | 44.12 | 20.69 | 61.96 | 44.77 | 61.3 | 62.01 | 49.14 |
> | PMC-Style Text-only Generation | 46.47 | 20.8 | 62.43 | 51.03 | 73.08 | 71.69 | 54.25 |
> | PMC-Style Text-image Generation | 48.82 | 20.4 | 63.38 | 51.08 | 73.08 | 72.06 | 54.8 |
> | Rubric Context-Aware Generation | 52.35 | 20.6 | 62.49 | 51.83 | 70.43 | 70.59 | 54.72 |
> | +Rubric Context-Aware Verification | 52.35 | 21.4 | 62.82 | 50.23 | 74.76 | 73.53 | 55.85 |
> | Qwen2.5-VL-7B-Instruct | 52.94 | 18.89 | 65.39 | 49.3 | 65.71 | 68.75 | 53.5 |
> | PMC-Style Text-only Generation | 54.71 | 22.15 | 63.89 | 53.23 | 67.07 | 65.44 | 54.41 |
> | PMC-Style Text-image Generation | 51.76 | 21.7 | 64.31 | 53.43 | 68.03 | 71.69 | 55.15 |
> | Rubric Context-Aware Generation | 58.24 | 23.5 | 65.41 | 53.83 | 68.03 | 75 | 57.33 |
> | +Rubric Context-Aware Verification | 57.06 | 23.15 | 66.36 | 53.78 | 67.79 | 77.21 | 57.56 |
>
>
> This decomposition shows that:
>
> * **The direct effect of adding images to a PMC-style generator is modest** (≈ +0.5-0.7 points on average).
> * **Rubric-guided generation and verification provide the majority of the remaining headroom**, especially for the stronger 7B student.
> * Verification consistently yields the best or tied-best average for both 3B and 7B, even though it operates on an already strong generator.
>
> We will update Section 4.2 to explicitly walk through this decomposition in the text, so that the role of each stage is clearer.
>
> ---
>
> **(b) Why rubric-based verification matters despite a ~1‑point average gain**
>
> We agree that the incremental average gain from "Rubric Context-Aware Generation" to "+ Verification" (≈ +1.1 for 3B and +0.2 for 7B) may look small at first glance. We would like to emphasize three points:
>
> 1. **Verification removes a large fraction of problematic items.**
>    From 23,635 generated candidates, 22,903 survive essential screening; after applying the Stage-2/3 rubric and a high threshold (τ=0.967), only **13,087** questions remain (≈45% are discarded).
>    Items rejected by the verifier typically:
>
>    * leak the diagnosis directly from the caption,
>    * are not self-contained,
>    * contain clinically inconsistent statements,
>    * can be answered without looking at the image.
>
>   Examples of these failure modes are documented in Appendix G.
>
> 2. **Verification helps most on clinically grounded benchmarks.**
>    Under a fixed 5k training size, adding verification:
>
>    * boosts **SLAKE** from 70.43 $\to$ 74.76 and **VQA‑RAD** from 70.59 $\to$ 73.53 for the 3B model, and
>    * improves **VQA‑RAD** from 75.00 $\to$ 77.21 for the 7B model.
>      These datasets require careful clinical and image-grounded reasoning, so the verifier's focus on self-containment, clinical validity, and image-text consistency is directly reflected there.
>
> 3. **RLVR is relatively robust to moderate label noise.**
>    Our training uses reinforcement learning with verifiable rewards (GRPO-style RLVR). Because RL optimizes expected reward rather than fitting each individual label, it can tolerate a certain amount of noisy supervision. As a result, **aggressively removing bad items improves safety, consistency, and auditability more than it boosts multiple-choice accuracy**, especially once the base dataset is already strong.
>
> Finally, beyond Table 2, **Table 5 shows that models trained with MedSynVQA outperform those trained on PMC‑VQA or m23k under the same RLVR framework** (e.g., for 7B: 53.66 (RL on PMC‑VQA) vs 57.56 (RL on MedSynVQA)), indicating that our rubric-guided generator-verifier pipeline improves **quality per training example**, not just scale or use of PMC figures.
>
> We will clarify and connect the qualitative examples (Appendix G) to the quantitative impact in Sec. 4.2.

---

> > ### Author Response · Authors · 2025-11-24
> >
> > ### 2. Main concern: lack of direct data quality evaluation
> >
> > We appreciate this important suggestion.
> >
> > **(a) Existing intrinsic quality analyses**
> >
> > While our main results focus on downstream VQA performance, the current manuscript already includes several intrinsic quality assessments:
> >
> > * **Decontamination analysis.**
> >   Appendix F performs systematic text and image decontamination between MedSynVQA and all evaluation benchmarks using MinHash + Levenshtein for text and hashing + embedding search for images. No exact or near-duplicate questions or images are found under our conservative thresholds, and overlap rates at high similarity cutoffs are <0.3%.
> >
> > * **Rubric statistics and rejection patterns.**
> >   Appendix D/E report pass rates for each rubric item (e.g., "Plausible Distractors," "Stem Self-contained," "Medical Inaccuracy") and show that many candidates fail specific gates such as forbidden terms, multiple keys, or medical inaccuracies.  This provides **quantitative evidence about dataset cleanliness** beyond downstream accuracy.
> >
> > * **Failure-case analyses.**
> >   Appendix G presents concrete examples that are rejected by the verifier but would otherwise enter the training set, illustrating how the rubric removes diagnosis leakage, unsupported clinical claims, and ambiguous stems.
> >
> > We will make these intrinsic analyses more visible in the main text (Section 3.3-3.4 and 4.2) and explicitly phrase them as **direct measurements of data quality**, not just implementation details.
> >
> > **(b) Human expert evaluation**
> >
> > We fully agree that a formal human study will further strengthen the paper. We are still conducting a **small-scale clinician evaluation** in which board-certified radiologists and pathologists rate a stratified sample of MedSynVQA questions. Please stay tuned for the update.
> >
> > ---
> >
> > We hope these clarifications and additional analyses address your concerns. We are grateful for your suggestions, which have helped us improve both the empirical analysis and the presentation of MedVLSynther and MedSynVQA.

---

> > > ### Comment · Reviewer_57hD · 2025-11-24
> > >
> > > Thank you very much for the timely reply. The more detailed ablation study has addressed most of my concerns.
> > >
> > > Overall, I am very positive about this paper. From my perspective, large-scale, automatic, high-quality data acquisition is the most critical factor and gap in building medical LMMs. I am encouraged to see a paper that makes such a substantial effort to thoroughly analyze the data acquisition process.
> > >
> > > I have raised my score. I believe the paper clearly meets the ICLR publication criteria. However, I cannot give it a higher score because it still relies on the idea of leveraging existing paper data to obtain more medical data. I believe we should continue exploring additional and more scalable diverse data sources.
> > >
> > > Thank you.

---

> > > > ### Author Response · Authors · 2025-11-24
> > > >
> > > > Thank you very much for your thoughtful review and for taking the time to re-evaluate our paper. We truly appreciate your positive assessment and are glad that the extended ablation study helped address your concerns. We fully agree that exploring additional, scalable, and diverse medical data sources is an important direction. Thank you again for your positive feedback and support of our submission.

---

> ### Author Response · Authors · 2025-12-04
>
> We have incorporated an **expert study** into the revised manuscript, now detailed in **Appendix G (Expert Evaluation Statistics)**, where five board-certified imaging specialists assess both verifier-accepted and verifier-rejected MedSynVQA items.
>
> To facilitate transparency and reproducibility, we also provide the full original questionnaire in the **supplementary ZIP file**.

---

### Author Response · Authors · 2025-11-24
**General Response**

Dear AC and SACs,

We provide a brief summary about our paper *MedVLSynther*, a scalable data pipeline that leverages generator–verifier LMMs to create a multimodal medical VQA training set. The significance of this work lies in democratizing access to high-quality training data and enabling the development of more robust and capable downstream medical VLMs via reinforcement learning with verifierable rewards. We believe this contribution can help catalyze future progress in multimodal medical AI.

We also want to express our deep appreciation for the reviewers’ thoughtful engagement during the discussion phase. After considering our rebuttal and revised manuscript, all reviewers provided positive assessments, **and the scores were raised to 6/6/6/8 (from the initial 2/4/6/6)** prior to the discovery of the information-leakage issue and the subsequent reversion to the original ratings.

In particular, Reviewer **57hD** wrote that they are *"very positive about this paper"* and that it *"clearly meets the ICLR publication criteria,"* while Reviewer **XqLm** noted that the new experiments *"successfully addressed the primary concerns regarding data efficiency and the necessity of multimodal generation,"* and Reviewer **n1Sq** stated that our clarifications *"address most of my concerns" and updated the review accordingly*. The remaining reviewer **7G9c** was already *supportive* from the outset.

Lastly, we are grateful to the reviewers for their careful and constructive engagement, and to the AC and SACs for considering the full discussion-phase evaluations and revised manuscript when making the final decision.

Sincerely,

Authors of paper #9539

---

### Meta-Review · Area_Chair_mvPX · 2026-01-13

**Summary:**

This submission introduces MedVLSynther, a rubric-guided generator–verifier pipeline that synthesizes multiple-choice medical VQA examples from open biomedical literature (figures, captions, and in-text references), producing MedVLSynther-13K . The work also demonstrates that training open-weight LMMs with reinforcement learning using verifiable rewards improves performance across six medical VQA benchmarks, with extensive ablations on pipeline components and dataset scale. This work is quite solid, and address one of bottle neck on AI for healthcare. The rebuttal is strong, addressed most of the concerns, although there are minor concerns on novelty, verification reliability and benchmark representativeness.

**Reviewer Concerns:**

Not outstanding remain concerns.

**Reviewer Scores:**

The rebuttal is strong, all the reviewers either remain high scores or increase the scores.

---

### Decision · Program_Chairs · 2026-01-26

Accept (Poster)